# Non-monotonic changes in Asian Water Towers' streamflow at increasing warming levels

Tong Cui [1,11], Yukun Li [1,11], Long Yang [2,3,11], Yi Nan [1], Kunbiao Li[1], Mahmut Tudaji[1], Hongchang Hu[1], Di Long [1], Muhammad Shahid[4], Ammara Mubeen[4,5], Zhihua He[6], Bin Yong[7], Hui Lu [8], Chao Li[9], Guangheng Ni[1], Chunhong Hu[10] & Fuqiang Tian [1] ✉

Previous projections show consistent increases in river flows of Asian Water Towers under future climate change. Here we find non-monotonic changes in river flows for seven major rivers originating from the Tibetan Plateau at the warming levels of 1.5 °C, 2.0 °C, and 3.0 °C based on an observation-constrained hydrological model. The annual mean streamflow for seven rivers at 1.5 °C warming level decreases by 0.1–3.2% relative to the present-day climate condition, and increases by 1.5–12% at 3.0 °C warming level. The shifting river flows for the Yellow, Yangtze, Brahmaputra, and Ganges are mostly influenced by projected increases in rainfall, but those for the Mekong, Salween, and Indus are dictated by the relative changes in rainfall, snowmelt and glacier melt. Reduced river flows in a moderately warmed climate threaten water security in riparian countries, while elevated flood risks are expected with further temperature increases over the Tibetan Plateau.

Large rivers, including the Yellow, Yangtze, Mekong, Salween, Brahmaputra, Ganges, Indus, and others, originating from the Asian Water Towers, including the Tibetan Plateau (TP) and its surrounding mountain ranges, provide valuable freshwater resources for more than 1.4 billion people living downstream[1,2]. Understanding how the river flows respond to future climate change is a pressing need for water security and the sustainable socioeconomic development of riparian countries. Previous studies project stable or increasing changes in river flows across major TP rivers under a warming climate, although the percentages of change vary considerably[3–9]. For instance, Lutz et al.[4] show that the total runoff of five TP river basins is expected to increase by 4.1–10.0% by 2041–2050 relative to the period 1998–2007. Wang et al.[9] found 1.0–7.2% increases in annual runoff over major TP rivers by the end of the 21st century. The magnitudes of changes are contrasted to Su et al.[5] which show more dramatic increases (i.e., 2.7–22.4%) in runoff by 2041–2070 relative to the period 1970–2000. Despite the inconsistent magnitudes of changes, the projected increasing tendency in river flows provides a stimulus for agricultural expansion in downstream riparian countries[10].

Here we find that the responses of river flow over major TP rivers, including the Yellow, Yangtze, Mekong, Salween, Brahmaputra, Ganges, and Indus (Supplementary Table 1), to different global

[1]State Key Laboratory of Hydroscience and Engineering, Department of Hydraulic Engineering, Tsinghua University, Beijing 100084, China. [2]School of Geography and Ocean Science, Nanjing University, Nanjing 210023, China. [3]Frontiers Science Center for Critical Earth Material Cycling, Nanjing University, Nanjing 210023, China. [4]Department of Civil Engineering, University of Engineering and Technology, Lahore 54890, Pakistan. [5]School of Civil and Environmental Engineering, National University of Sciences and Technology, Islamabad, Pakistan. [6]Centre for Hydrology, University of Saskatchewan, Saskatoon, Saskatchewan S7H, Canada. [7]State Key Laboratory of Hydrology-Water Resources and Hydraulic Engineering, Hohai University, Nanjing 210098, China. [8]Department of Earth System Science, Ministry of Education Key Laboratory for Earth System Modeling, Institute for Global Change Studies, Tsinghua University, Beijing 100084, China. [9]Key Laboratory of Geographic Information Science, Ministry of Education, East China Normal University, Shanghai 200241, China. [10]China Institute of Water Resources and Hydropower Research, Beijing 100038, China. [11]These authors contributed equally: Tong Cui, Yukun Li, Long Yang. ✉e-mail: tianfq@tsinghua.edu.cn

warming levels (i.e., 1.5 °C, 2.0 °C, and 3 °C) are divergent based on observation-constrained hydrological simulations and state-of-art climate projections. The annual mean streamflow for the seven rivers at the warming level of 1.5 °C decreases by 0.1–3.2% relative to the present-day climate condition. The decreasing tendencies maintain for the Mekong, Salween, Brahmaputra, and Indus at 2.0 °C, but reverse to increasing for the Yellow, Yangtze, and Ganges (i.e., 0.5–2.2%). All seven rivers tend to experience increased annual mean streamflow (ranging from 1.5 to 12%) at the warming level of 3.0 °C. Our analysis relies on a physically-based, spatially-distributed hydrological model, the Tsinghua Representative Elementary Watershed (THREW) model[11]. The THREW model is incorporated with modules characterizing different runoff-generation processes, and is extensively calibrated for individual runoff-generation modules (e.g., rainfall-runoff, snowpack dynamics, and glacier evolution) against observations over TP ("Methods", Supplementary Table 2 and Supplementary Table 3). We adopt the WATCH Forcing Dataset (WFD) as the meteorological forcing for model calibration and validation. The observation-constrained THREW model is then driven by the output from 22 Coupled Model Intercomparison Project Phase 6 (CMIP6)[13] models (Supplementary Table 4) under the historical period (i.e., 1985–2014, HIST) as the baseline simulation and under two Shared Socioeconomic Pathways (SSPs) scenarios, i.e., SSP 2-4.5 (SSP245) and SSP 5-8.5 (SSP585), as future hydrological simulations (FUTU, see "Methods"). The average results of the SSP245 and SSP585 scenarios are used to represent the future simulation due to slight variations between the two scenarios. Differences in annual or seasonal mean streamflow between FUTU and HIST simulations, represented as (FUTU-HIST)/HIST, are used to quantify the magnitudes of changes in river flows over TP. The 22 CMIP6 models are selected based on their

performance in reproducing present-day climate conditions over TP[14,15]. We interpolate the 22 CMIP6 model outputs from various spatial resolutions into 0.5° (i.e., same as WFD) through a bilinear interpolation scheme. The biases in the interpolated model output are further corrected against WFD based on a multiplicative bias-correction approach[16] (see "Methods", Supplementary Table 5). We focus on the warming levels of 1.5 °C and 2.0 °C, because they represent the central aim of the Paris Agreement, which is to keep the global temperature rise below 2.0 °C[17]. We choose the warming level of 3.0 °C as a "worst-case" scenario, since it is the projected warming level by the end of the 21st century given current nationally-determined mitigation ambitions[18]. The three warming levels are attained around the 2030 s, 2050 s, and 2070 s, respectively, in the 22 CMIP6 models (Supplementary Table 6).

## Results
### Changes in annual and seasonal mean streamflow
Annual mean streamflow over TP is expected to increase at the warming level of 3 °C, although the magnitudes of change vary considerably across the seven rivers (Fig. 1). Annual mean streamflow over the Yangtze shows increases of 12.0 ± 24% by 2070 s relative to the historical period (i.e., 1985–2014). The magnitudes of change are comparable across the Brahmaputra (7.3 ± 11%), Yellow (8.3 ± 14%), and Ganges (9.5 ± 11%), but are less significant for the Salween (1.9 ± 5%), Mekong (2.6 ± 7%), and Indus (1.5 ± 9%) (Supplementary Dataset 1). The increasing tendencies in annual mean streamflow across the seven rivers are overall conserved in seasonal mean streamflow except for the Salween, Mekong, and Indus. The three rivers experience decreased summer streamflow (i.e., June to August), which is accompanied by notable increases in autumn streamflow (i.e., September to

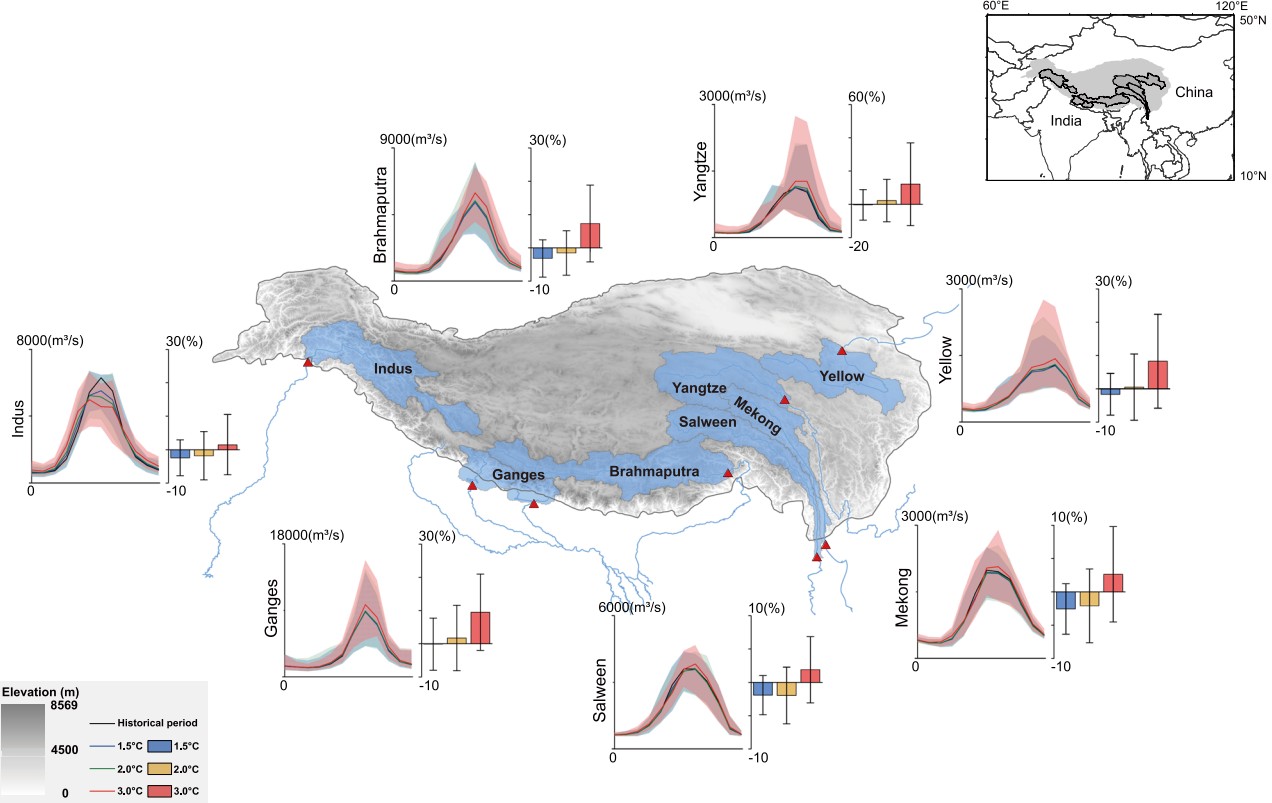

**Fig. 1 | Relative changes of river flows at the warming levels of 1.5 °C, 2.0 °C, and 3.0 °C for the seven rivers.** Color bars represent the ensemble mean from the hydrological simulations driven by the output from 22 CMIP6 models, while error bars represent one standard deviation. Solid lines represent the ensemble mean, while shadings show the range. Red triangles show basin outlets. Gray shading represents elevation (in meters), while the blue shading shows the basin boundary. The Tibetan Plateau boundary is defined by the elevation contour of 2500 m[65].

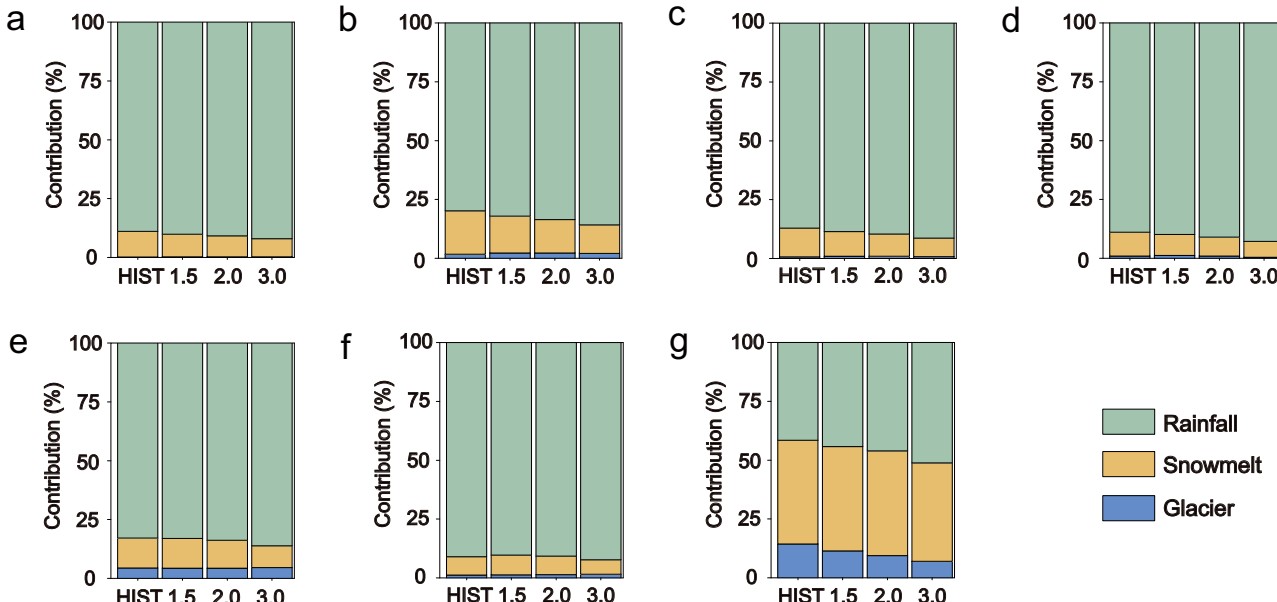

**Fig. 2 | Relative contribution of different runoff components to total runoff at the warming levels of 1.5 °C, 2.0 °C, and 3.0 °C for the seven basins. a** Yellow; **b** Yangtze; **c** Mekong; **d** Salween; **e** Brahmaputra; **f** Ganges; **g** Indus. The first column shows the contribution during the historical period (1985–2014, HIST), while the second to the fourth columns show the contribution at the global warming level of 1.5 °C, 2.0 °C, and 3.0 °C, respectively.

November, Supplementary Fig. 1). Uneven magnitudes of changes in streamflow of different seasons lead to temporal shifts of peak streamflow timing for these rivers. We observe a delayed occurrence of peak streamflow by one month (i.e., from July to August) for the Salween and Mekong, while the peak streamflow is advanced by one month (i.e., from July to June) for the Indus (Fig. 1). The occurrence of peak streamflow remains unchanged for the Yangtze, Yellow, Ganges, and Brahmaputra.

The consistent increasing tendencies in annual mean streamflow for the seven rivers become divergent at the warming level of 2.0 °C (Fig. 1). Hydrological simulations show decreased annual streamflow (i.e., contributed by the decreased streamflow in summer and autumn) over the Mekong, Salween, Brahmaputra, and Indus by 2050 s. The direction of changes in annual mean streamflow maintains for the Yellow, Yangtze, and Ganges between the warming levels of 2.0 °C and 3.0 °C, although the magnitudes are much smaller under the warming level of 2.0 °C. For instance, the changes of annual mean streamflow are 2.2 ± 13% and 1.7 ± 10% for the Yangtze and Ganges, respectively, no more than one-fifth of the magnitudes of change at the warming level of 3.0 °C. The changes for the Yellow, Yangtze, and Ganges in a moderately warming climate (i.e., 2.0 °C) are broadly consistent with previous studies that show increased streamflow across major TP rivers by 2050 s and beyond[3–5,9].

However, the consistently increasing tendencies in annual mean streamflow at the warming level of 3.0 °C are completely reversed when the increase of global mean temperature attains 1.5 °C (Fig. 1). All seven rivers show decreased annual mean streamflow relative to the historical period, with the change magnitudes ranging from 0.1 ± 8% to 3.2 ± 5%. The Mekong, Brahmaputra, and Indus, among others, demonstrate the most notable decreases, while the changes are relatively small for the other rivers, i.e., less than 0.5% for the Yangtze and Ganges. The peak streamflow timing for the seven rivers under a moderately warming climate (i.e., at the warming level of 1.5 °C and 2.0 °C) is overall consistent with that during the historical period. The sharp contrasts in annual and seasonal mean streamflow for the seven rivers between the warming level of 1.5 °C and 3.0 °C, together with the divergent change directions at 2.0 °C, highlight non-monotonic responses of the hydrosphere to anthropogenic

climate warming over TP (particularly along its southern and eastern rims).

## Contributions of different runoff components

The non-monotonic changes in river flow highlight complexities in runoff-generation processes over TP[19,20]. We further look into the changes of different runoff components at three warming levels for each drainage basin, to highlight the physical processes that lead to the changing river flows. Based on the physiographic and climatic environment of TP, runoff-generation processes are classified into three categories, i.e., rainfall runoff (including rain-on-snow/ice and the mixture of snow/rain), snowmelt runoff, and glacier runoff[4,21]. The three runoff components are represented in the THREW model by individual modules (see definition of different runoff components in "Methods"). The sum of the three components constitutes the total runoff for each basin.

Rainfall runoff predominantly contributes (i.e., 79.8 ± 1–91.0 ± 1%) to total runoff across the seven basins except for the Indus, followed by snowmelt (10.8 ± 1–18.4 ± 1%) and glacier melt (0.2 ± 0.01–4.4 ± 0.2%, Fig. 2). This is mainly because hydrological regimes for the six basins are dominated by either the East Asian monsoon (i.e., for the Yangtze and Yellow) or the Indian monsoon (i.e., for the Ganges, Salween, Brahmaputra, and Mekong)[22–24]. The monsoonal rainfall from May to October accounts for ~90% of annual rainfall on average for the six basins (Supplementary Fig. 2). The Indus has the largest snow cover (i.e., 39% of the drainage area) of the seven basins, with total runoff contributed the most by snowmelt, followed by rainfall and glacier melt.

Although rainfall dominates total runoff for the seven rivers except for the Indus, changes in river flows are determined by relative changes of the three runoff components rather than by rainfall alone. For instance, annual total precipitation is projected to increase slightly (i.e., up to 2.5%) for the Salween and Mekong at the warming levels of 1.5 °C and 2.0 °C (Supplementary Fig. 3). Meanwhile, the increasing temperature leads to the depletion of snow accumulation (Supplementary Fig. 4 and Supplementary Fig 5), and thus decreased runoff from snowmelt. This is demonstrated by the fact that at 1.5 °C and 2.0 °C warming levels, the snow cover in the upper Mekong and

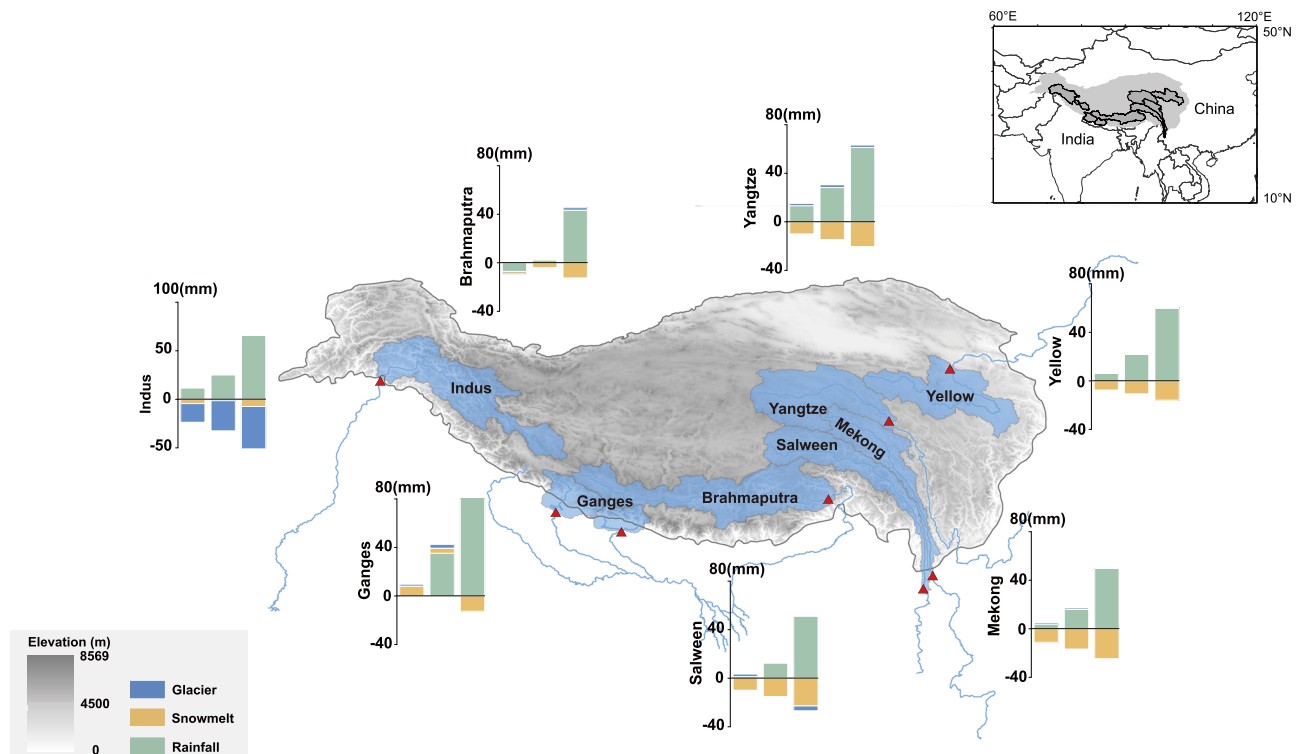

**Fig. 3 | Relative changes in annual mean rainfall runoff, snowmelt runoff, and glacier runoff at the warming levels of 1.5 °C, 2.0 °C, and 3.0 °C for the seven basins.** The first to third bars show the changes at the warming level of 1.5 °C, 2.0 °C, and 3.0 °C, respectively. Red triangles show basin outlets. Gray shading represents elevation (in meters), while the blue shading shows the basin boundary. The Tibetan Plateau boundary is defined by the elevation contour of 2500 m[65].

Salween is projected to decrease by 11.76% −21.87%. Decreased snow cover leads to a negative contribution of snowmelt to total runoff for the Salween (−1.6 ± 0.5%) and Mekong (−2.2 ± 0.4%). The magnitudes of change in snowmelt outweigh the contribution of increased rainfall to total runoff, ultimately resulting in decreased river flows for the Mekong and Salween (Fig. 3). However, when the increase of global mean temperature attains 3.0 °C, the fraction of precipitation that falls as snow is considerably decreased (Supplementary Fig. 6 and Supplementary Fig. 7). More precipitation falls as rain, resulting into more rainfall runoff. The contribution of increased rainfall outweighs that of decreased snowmelt and subsequently leads to increased river flows. The relative changes of snowmelt and rainfall runoff collectively determine both the direction and magnitudes of changes in river flow for the Yellow, Yangtze, Brahmaputra, and Ganges (in addition to the Mekong and Salween). For the Mekong and Salween, the earlier depletion of accumulated snow (as evidenced by the reduced snow coverage in July, Supplementary Fig 5) leads to the dominance of rainfall in runoff generation, and thus delayed peak streamflow at the warming level of 3 °C.

For the Indus, changes in total runoff are collectively determined by the projected changes in rainfall and snow/glacier melt. The magnitudes of changes in snowmelt remain largely unchanged (i.e., 41.9−44.1%), while rainfall and glacier melt vary considerably across different warming levels (Fig. 3). The decreased glacier melt is mainly associated with the glacier shrinkage induced by the increased temperature and thus more energy for melting. For instance, the spatial coverage of glaciers decreased from 6.0% during the historical period to 2.5% at the warming level of 3.0 °C. The reduced snow and ice converted from rainfall (due to increased temperature) are also responsible for the reduced glacier melt runoff over the Indus. Earlier snowmelt and glacier under the warming level of 3.0 °C leads to advanced peak streamflow from July to June for the Indus (Supplementary Fig. 5).

Evapotranspiration, an additional determinant of the basin-scale water balance, shows consistent increases across the seven basins under different warming levels (Supplementary Fig. 8). However, the magnitudes of changes in evapotranspiration are relatively small (i.e., less than 3%) during the dominant runoff-generating seasons (i.e., June to August). Changes in total runoff are thus predominantly dictated by the projected changes in rainfall for the seven basins, with changes in the timing and extent of snow/glaciers additionally important for the Mekong, Salween, and Indus. Here we adopt the multi-model ensemble means of 22 GCMs output as the rainfall projection over TP. This is mainly because the ensemble means are more consistent with observations than any single model during the historical period[5]. Despite this, we notice a wide range of rainfall projections for individual basins. For instance, the annual average precipitation is projected to change by −4.8% to 33.2% over Yellow and −3.6% to 13.7% over Mekong river basin at the warming level of 3.0 °C. Our results highlight the importance of projecting the spatial and temporal variability of rainfall over TP under a warming climate.

## Discussion

The projected decreases in river flow in a moderately warming climate (i.e., at the warming level of 1.5 °C and 2.0 °C) pose threats of water scarcity to riparian countries with booming populations and agricultural expansion. For instance, the population in Pakistan, India, Bangladesh, and Nepal is projected to increase by 21.6%, 16.3%, 10.5%, and 15.7% by 2040s[25]. The current population under water scarcity[26] (with larger water consumption than water availability, evaluated by water scarcity index, see "Methods") is approximately 428 million. Take the Indus for example, the upper Indus provides a quarter of the total water supply for the downstream countries (mainly Pakistan and India)[27]. Insufficient surface water supplies already incurred severe depletion of the aquifers over the region[28]. A 2.5% decrease in river flows for the Indus by the 2030 s will lead to a 2.6% increase in

population with insufficient water resources (Supplementary Fig. 9), and most likely intensify the exploitation of deep aquifers. The pressing need for water resources also exists for the Mekong. According to the Mekong River Commission Council, the total irrigation area in the Mekong River Basin (mainly its downstream portion) will be nearly doubled by 2040s[29]. This exerts an intensified water demand on the rivers originating from TP. It should be noted that the conditions of water scarcity are also closely tied to other factors, including changes in lifestyle, infrastructural development, and governmental policies. The index adopted here provides a simple measure of water scarcity from the supply side.

While the projected increases in streamflow at the warming level of 3.0 °C bode well for water resources of riparian countries, the potential of intensified flood hazards (i.e., associated with either extreme rainfall flooding and/or outburst floods related to glacier retreat under anthropogenic warming) highlight the needs for improved flood risk mangement[30,31]. Changes in the occurrence of peak streamflow (i.e., for the Mekong, Salween, and Indus) also highlight the importance of improved planning of agricultural activities for more efficient water utility under a warming climate[32]. Understanding complex hydrologic responses to climate change for the Asian Water Tower is a pivotal step toward effective water resources planning and management. Our state-of-art projection of future river flows over TP offers a timely reference to stakeholders and decision-makers of riparian countries that are experiencing notable socioeconomic transitions.

## Methods

### Datasets

Time series of observed daily streamflow at the stream gauging stations of the Yellow, Yangtze, Mekong, Salween, Brahmaputra, Ganges, and Indus are provided by the Hydrology Bureau of China, Department of Hydrology and Meteorology of Nepal, and the Pakistan Water and Power Development Authority. The periods of the streamflow observations span from 1980 to 2012 (see Supplementary Table 1 for details). The observed streamflow series are used to calibrate and validate the hydrological model (see details below).

We extract meteorological variables (i.e., precipitation, temperature, and potential evapotranspiration) from the WATCH Forcing Data (WFD) for hydrological model calibration and validation[12]. The temporal resolution of the WFD is daily, and the spatial resolution is 0.5°. Since WFD performs relatively better than other datasets over TP[33–35], it is also used as the reference for bias-correcting climate model outputs in reproducing historical climate conditions (see ref. [36] for example). Due to complex terrains and insufficient in-situ rain gauges, there is a large underestimation of precipitation for the WFD over the Indus. We then multiply the daily precipitation for the Indus by 1.61. The multiplicative factor is determined based on basin-scale water budget analysis (similarly see refs. [37–39]). Other datasets used in hydrological modeling include topography, land use/land cover, soil properties, snow cover, glacier extent, ice thickness, and glacier mass balance, and are summarized in Supplementary Table 2.

We obtain daily temperature and precipitation for 1850–2014 and future climate projections under two SSPs scenarios, i.e., SSP245 and SSP585 generated by 22 CMIP6 models[13,40] (Supplementary Table 4). The performance of the 22 CMIP6 models over TP has been evaluated in our study (see ref. [15]). We interpolate the CMIP6 model output from variable spatial resolutions to 0.5° (i.e., consistent with WFD) based on a bilinear interpolation scheme. We additionally compute the daily potential evapotranspiration based on Van Pelt et al.[41] (see Supplementary Methods) for all the 22 CMIP6 models under the two SSPs scenarios.

### Bias correction of climate forcing

We apply the N-dimensional probability density function transform of the multivariate bias correction (MBCn) approach to the interpolated CMIP6 model output[16]. The MBCn approach considers the multivariate dependence structure of the climate model output. It shows substantial improvement compared to those univariate bias-correction algorithms or other multivariate bias-correction algorithms[42]. Here the MBCn approach is first trained based on comparisons between the CMIP6 model outputs (i.e., the historical simulation) against WFD during the period 1980–2000. We then test the performance of the bias correction approach for precipitation and temperature during the period 2001–2014. The seasonal cycle of basin-average precipitation and temperature from the bias-corrected ensemble-mean outputs of the 22 CMIP6 models, as well as their annual mean values, match well against the WFD (Supplementary Table 5 and Supplementary Fig. 10). The bias-corrected CMIP6 model outputs are then used to provide meteorological forcing for hydrological simulations during both the historical period and in future climates.

### Determination of warming levels

We consider three warming levels, i.e., 1.5 °C, 2.0 °C, and 3.0 °C relative to the pre-industrial period (1850–1900). The warming levels of 1.5 °C and 2.0 °C are within the targets of the Paris Agreement[17], while 3.0 °C represents the projected warming under current nationally-determined mitigation goals[18]. We calculate the global mean temperature within a 30-year running period starting from the year 2015. We determine the warming period if the 30-year running mean global surface temperature for the first time exceeds the targeted warming level. This is also known as the time-sampling method, and is recommended by the Inter-Sectoral Impact Model Intercomparison Project (ISIMIP)[43,44]. Choosing a fixed period for intercomparisons among all climate models might be not desirable. This is because the intrinsic contrasts among the 22 CMIP6 models lead to diverse trajectories in global mean temperature within a fixed period. For instance, the UKESM1-0-LL model projects a 3.0 °C increase in global mean temperature by 2052 under SSP245, while the temperature increase is 2.0 °C for the EC-Earth3-Veg model by then (Supplementary Table 6). The 22 CMIP6 models attain the targeted warming levels around the 2030 s, 2050 s, and 2070 s, respectively. As there are no significant differences between the two SSPs scenarios, i.e., SSP245 and SSP585, we calculate the average of the two scenarios to represent future climate conditions under specific global warming levels.

### Hydrological model

We carry out hydrological simulations for both the historical period and the period under different global warming levels based on the Tsinghua Representative Elementary Watershed (THREW) model. The THREW model is a physically-based, spatially-distributed hydrological model[11]. The model solves the mass, momentum, and energy balance equations by discretizing the entire basin into a collection of Representative Elementary Watersheds (REWs). The REWs are characterized based on their physiographic properties within them. The THREW model has been successfully applied to many river basins across China and the United States[45–48], with good model performance achieved.

We apply a modified version of THREW in the present study. The modified THREW is equipped with modules for glacier evolution and snowpack simulation (see ref. [11] and ref. [49] for details). It incorporates a dynamic Glacier-REW (GREW) scheme to simulate the transient response of glacier changes. Each REW is first divided into elevation bands with an equal increment of 200 meters. The region covered with glaciers along each elevation band is then regarded as a GREW. Changes in both the glacier extent and thickness are simulated over each GREW instead of the entire REW[50]. We adopt the degree-day method proposed by Hock et al. to simulate the glacier melt rate. The mass balance with the volume-area scaling relation introduced by Chen and Ohmura[51] is adopted to represent glacier advance or retreat. We adopt an algorithm proposed by Luo et al.[49] for the accumulation

of glacier mass. Compared to other glacier hydrological models, the simulation of glacier volume-area changes and glacier mass balance can reflect the transient response of the glacier to the changing climate. We do not claim the superiority of the glacier module in the THREW model over other widely used glacier models, most of which focus on fine-scale simulation of glacier dynamics for specific glaciers instead of basin-scale water budget. In this study, we enhance the applicability of the THREW model, to capture the key features of basin-wide water balance (contributed by glacier evolution, snowpack, and rainfall) over TP through extensive model calibration and validation (see details below). We refer the readers to Marzeion et al.[50] for a comprehensive review of existing glacier models. Datasets required for THREW are summarized in Supplementary Table 2, with more details about the model physics provided in Supplementary Method.

We categorize runoff-generation processes from the THREW model into three different components, i.e., rainfall runoff, snowmelt runoff, and glacier runoff. Rainfall runoff includes runoff that is generated from rainfall over the entire basin (either over snow/glacier coverage or not). Snowmelt runoff includes runoff that is generated due to snowmelt over snow coverages, while glacier runoff is defined as runoff contributed by glacier melt or snowmelt over the glacier coverage (by following Equations 2–12 in the Supplementary Method).

## Model calibration and validation

We calibrate the THREW model using the WFD as meteorological forcing during 1980–2012. Records of observed streamflow are split into calibration and validation periods (Supplementary Table 3). The simulated daily streamflows for the seven rivers are compared against the observed streamflow at the corresponding stream gauging station. Here we adopt the pySOT, a multi-objective model calibration approach[52], to automatically determine the optimal values of model parameters. All the parameters (including their physical meaning and possible range) are summarized in Supplementary Table 7. The Nash Sutcliffe efficiency coefficient (NSE)[53] and correlation coefficient are used as the objective functions for streamflow simulation. We additionally calculate the correlation coefficient to verify the model's performance in capturing the dynamics of glacier mass balance for each REW over the Indus (Supplementary Table 8). The optimal values of model parameters are listed in Supplementary Table 9.

Unlike other studies that only focus on the model's output at basin outlets (refs. [4–8]), we spent additional efforts in tuning parameters that determine the performance of snowpack and glacier evolution module. We calibrate the parameters (including the degree-day factor, and the coefficient in the snow depletion curve) of the snowpack module by comparing the fraction of simulated snow coverage against the observed snow cover over the entire basin. The observed snow cover over TP is provided by the long-term TP daily 5-km cloud-free snow cover extent record (TPSCE)[54]. We calibrate the parameters (including the degree-day factor, and the parameters in the glacier volume-area scaling relationship) of the glacier evolution module by comparing the simulated glacier spatial extent and ice thickness over each REW against various observation-based products. We adopt the First Chinese Glacier Inventory (CGI-1)[55] and Second Chinese Glacier Inventory for the glacier extent over the regions within China (SCGI)[56] as well as the Randolph Glacier Inventory Version 6.0 (RGI6.0)[57] for the regions outside China. The initial glacier extent for historical simulation is determined by CGI-1 and RGI6.0 (Supplementary Method). The initial glacier thickness is computed based on an area-volume scaling relationship and is constrained by referring to Millan et al.[58].

The calibrated THREW model can capture key features of daily and monthly streamflow for all seven river basins. The mean value of the Nash Sutcliffe efficiency coefficient is 0.78 for daily streamflow and 0.85 for monthly streamflow during both the calibration and validation period (Supplementary Table 3 and Supplementary Fig. 11). The

correlation coefficients exceed 0.90 for both the daily streamflow and the monthly snow cover for the seven basins (Supplementary Fig. 12). The integration of simulated glacier extent for each REW also agrees well with the observed glacier extent provided by SCGI and RGI 6.0, with high correlation coefficients (>0.96, Supplementary Fig. 13).

We additionally compare the simulated tendency of glacier mass balance over the Brahmaputra and Indus against remotely-sense observations (from World Glacier Monitoring Service (WGMS)[59] and Hugonnet et al.[60]) during the period 2000-2011 (Supplementary Table 10). The correlation coefficient for the tendency of glacier mass balance between our simulation and Hugonnet et al. is 0.73 ($P = 0.05$). Our simulation also is consistent with WGMS, with a correlation coefficient of 0.31. The mean change rate of all REWs for the Indus is −0.42 m/yr from the simulation. The mean change rate is relatively larger than either WGMS (i.e., −0.22 m/yr) or Hugonnet et al. (2021) (i.e., −0.04 m/yr). This is expected as we notice that the two remotely-based observations demonstrate variations themselves in either the basin-wide change rates or over each REW. The presented simulation bias can be either related to the uncertainty in remotely-based observations or our model simulation (particularly the glacier module). We carry out sensitivity analyses for two important glacier module parameters, i.e., the annual glacier changes rate and degree-day factors, over the Indus (Supplementary Method). This is because the Indus has the largest contribution of glacier runoff to total runoff among all seven river basins. Results show the changes in parameters do not influence our conclusion in terms of non-monotonic changes in total runoff at different warming levels (Supplementary Table 11). This indicates that the glacier probably plays a minor role in the uncertainty.

## Water scarcity index

We adopt the water scarcity index to evaluate the condition of water demand relative to water availability. The index is defined as the ratio of total water withdrawn for all types of anthropogenic activities (including domestic, agricultural, and industrial sectors, see refs. [26,61,62]) to the total amount of water that is available for a specific region. The Inter-Sectoral Impact Model Intercomparison Project (ISIMIP2a) provides monthly water withdraws for agricultural, domestic, and industrial activities at the grid scale of 0.5°[63]. The simulated daily runoff is also provided by ISIMIP2a for the estimation of water availability. The water withdraws are estimated based on agricultural irrigation intensity, population, and socioeconomic development. The total amount of water that is available is 60% of river flows. The rest 40% of river flow is maintained as environmental flows[26]. A grid is regarded as water scarcity if the value of water scarcity index exceeds 1.0. We then estimate the total population that is under water scarcity for each river basin by counting the population over the grids with the water scarcity index larger than 1.0. The dataset of population is provided by the Gridded Population of the World collection (GPW)[64] at a spatial resolution of 0.5°.

To project water scarcity under global warming scenarios, we first estimate future global population growth at the grid level. The future population at each grid is estimated by multiplying the current population (use the year 2010 as the reference i.e., obtained from the GPW dataset) by the population growth rate (use the year 2030, 2050, and 2070, respectively) for the countries that contain the grid. The population growth rates for different countries are provided in ref. [25]. We assume that the spatial pattern of population growth and water withdrawal density remain unchanged. We estimate future water withdrawal under different warming levels by multiplying the historical water withdrawal over the grids by the percentage change of grid-based population. We estimate future water availability by multiplying the historical runoff over the grids within each basin by the percentage change of annual mean streamflow of the corresponding basin under different warming levels. The population under water scarcity at global

warming levels is then calculated as the total population over the grids with a future water scarcity index larger than 1.0 (see Supplementary Fig. 9 for details). The most pronounced changes in the total population under water scarcity are projected for the Indus and Ganges. The percentages of increase in the two basins are 2.5%, and 3.9% at the warming level of 1.5 °C and 2.0 °C, respectively.

## Data availability
The MERIT DEM is available at http://hydro.iis.u-tokyo.ac.jp/~yamadai/MERIT_DEM/. The NDVI is available at https://www.ncei.noaa.gov/products/climate-data-records/normalized-difference-vegetation-index.The LAI dataset is available at https://www.ncei.noaa.gov/products/climate-data-records/leaf-area-index-and-fapar. Department of Hydrology and Meteorology in Nepal for the Ganges is available at https://www.dhm.gov.np/request-data, and the Pakistan Water and Power Development Authority for the Indus are available at http://www.wapda.gov.pk. Soil hydraulic parameters are available at http://globalchange.bnu.edu.cn/research. The Watch Forcing dataset (WFD) is obtained from https://rda.ucar.edu/datasets/. The CMIP6 model outputs are available at https://esgf-node.llnl.gov/search/cmip6/. The RGI data is available at http://www.glims.org/RGI/randolph60.html.The ice thickness datasets are available at https://doi.org/10.6096/1007. The glacier mass balance data are available at https://wgms.ch/data_databaseversions and https://www.sedoo.fr/theia-publication-products/?uuid=c428c5b9-df8f-4f86-9b75-e04c778e29b9. H08 data is available at https://www.isimip.org/. The World Gridded Population is available at https://sedac.ciesin.columbia.edu/data/collection/gpw-v4. The results of this study are available as a supplement to this manuscript (Supplementary Dataset 1).

## Code availability
The hydrological model used to generate the results reported in this study is available from the corresponding author upon request. The MBC package used for bias correction in this study is available at https://cran.r-project.org/package=MBC.

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

## Acknowledgements

This study is supported by National Natural Science Foundation of China (Grant Nos. 92047301, 51825902, 51961125204, and 52109023), and the State Key Laboratory of Hydroscience and Engineering (Grant Nos. 2022-KY-03). The Tibetan Plateau boundary is provided by National Tibetan Plateau Data Center (http://data.tpdc.ac.cn). L.Y. also acknowledges support from the Fundamental Research Funds for the Central Universities (0209-14380804) and the Frontiers Science Center for Critical Earth Material Cycling Fund.

## Author contributions

F.T. conceived the idea and designed the research. T.C., Y.L., and L.Y. constructed the hydrological model, analyzed the data, and drafted the manuscript. Y.N., K.L., M.T., H.H., and D.L. contributed to the construction of the hydrological model. M.S., A.M., Z.H., and B.Y. contributed to glacier modeling. C.L. contributed to bias correction. H.L. and G.N. helped interpret the results. C.H. contributed to the ideas of the research and the interpretation of the results. All authors contributed to the discussion of the results.

## Competing interests

The authors declare no competing interests.
