## [Peer Review File · Nature Communications]

Non-monotonic Changes in Streamflow over Asian Water Towers at 1.5 °C, 2.0 °C and 3.0 °C Warming LevelsReviewer #1 (Remarks to the Author):

The Asian water tower is one of the most sensitive areas of climate change in the world and is also the source of many great rivers. The change of water resources in its upstream will affect the living of its downstream residents. In the paper, based on the observation-constrained hydrological model, the Tsinghua Representative Elementary Watershed (THREW) model, non-monotonic changes of river flows for seven river basins, under the warming levels of 1.5, 2.0, and 3.0 °C, are found. Firstly, the characteristics of streamflow under different warming levels were described respectively. Then, the contributions of different components were analyzed. On this basis, the influence on downstream is further discussed. This study has implications for water management in Asian water towers in a warming world. My overall comments for the current paper are as follows:

Abstract

1. It would be better to specify the value of change in Indus.
2. Line 57, "different global warming levels (i.e., 1.5 °C, 2.0 °C and 3 °C)" how to select the warming levels? Any reasons?
3. Line 58, "This is mainly associated with complex runoff components in TP river basins that respond differently as climate warms". The sentence seems to make no sense here. I noticed a similar sentence in line 109, which implies that contributions of different runoff components are about to be analyzed.
4. Line 66, "from 22 Coupled Model Intercomparison", why 22 CMIP6 models are selected? How to deal with the resolution with different CMIP6 models?

Results

5. We usually consider the summer season as June-August. What is the author's consideration in identifying summer as May-July here? Therefore, the period of autumn has changed too.
6. Line 85, "The declined streamflow in summer is accompanied by significant increases in autumn streamflow (i.e., August to October) in the two river basins. The sharp contrasts in summer and autumn streamflow correspond to a shift of peak monthly streamflow from July to August by 2070s for the upper Salween and Mekong". The phenomenon of "significant increases in autumn" is not only found in the two basins, but also in other basins, such as UYE, UYA and so on in supplementary fig.1, or even more significant. Besides, the "peak monthly streamflow" is obscure to me because I didn't draw the conclusion from the mentioned figs.
7. Line 89, "... over Salween and Mekong further contributes to negative ...". I would suggest deleting the "further" here as it seems to confuse the relationship of warming at 2°C and 3°C.
8. Line 101, "... across TP river basins except for the upper Indus". I would suggest stating the value of Indus.
9. Line 113, "There are three runoff components over TP, i.e., rainfall runoff (including rain-on-snow processes), snowmelt runoff, and glacier-melt runoff". I would like to know why there are three components contributing to streamflow. And evapotranspiration is mentioned at the end of the paragraph. Is there any connection between them?
10. Line 124, "For the warming levels of 1.5 °C and 2.0 °C, the decreases in snowmelt runoff outweigh increased rainfall runoff especially for the upper Salween and Mekong (Fig. 3)". The color bar in fig. 3 does not seem to agree with what the author says for the Salween. And there is an extra UM in fig. 3.
11. Line 141, "... consistent increases with temperature, ranging from 3.5% to 7.3%, ...". Does the range correspond to the change in temperature from 1.5 °C to 3 °C? Please specify it.

Implications

12. Line 157, "... and peak monthly streamflow (e.g., Fig. 2 for Mekong river basin) ...". As mentioned in 4, I do not understand how to understand the peak here.

Reviewer #2 (Remarks to the Author):

This paper describes a comprehensive hydrological modelling study to assess the response of large streams originating from the Tibetan Plateau to three different atmospheric warming levels. The authors have assembled a large amount of data sets and model approaches to provide streamflow projections and analyse the changes in water runoff from the considered headwater catchments from individual components (rain, snow melt, glacier melt). Particular attention is devoted to the calibration and validation of the components of the hydrological response to climate change and a complex pattern of runoff shifts is found, depending on the amount of atmospheric warming and the characteristics of the basin (e.g. percentage of glacier cover). Also, the effect of future changes in streamflow on water scarcity on the population in the regions surrounding the Tibetan Plateau is assessed.

Although it is obvious that the effort for conducting the study and setting up this comprehensive model framework was tremendous and the results are potentially interesting, I have detected a number of important weaknesses and problems in the present version of the study that – in my opinion – preclude publication in Nature Communications at the moment. These major issues are summarized below, followed by some more specific remarks. My review puts an emphasis on the representation of glaciers and their hydrological contribution.

Substantive comments:

- General presentation: The paper is generally well written, although at several places the language would benefit from a revision by a native English speaker. It is however most problematic that various important aspects of the methodology are only described superficially. Even when going to the Supplementary Material, many aspects of model implementation and/or decisions in setting up the modelling framework remain unclear. Model variables are unexplained or, at least, it cannot be tracked where they are taken from. Many acronyms are not defined which makes the reading more difficult. There are also loose ends: For example, in the main text a validation of model results with glacier mass balance data is promised but nowhere a result is shown – the reader is just expected to believe that the analysis was favourable.

- Referencing: Although the main text only allows a limited number of references, the choice might be revised to be broader and take into account more non-Chinese studies performed in the region. In my opinion, several relevant references are missing many data sets (Supplementary material) are present without providing a reference at all.

- Data sets: At several places the motivation for use of specific data sets was difficult to follow by me as little explanation is given. For example, the study strongly relies on the WFD data set for bias correcting GCM data. Some more information is needed why this data is deemed optimal for the region, even more as the authors have not underlain the use of the data set with a publication documenting its applicability. On the glacier side, there are various widely accepted large-scale data sets that have been used by basically all previous recent studies in the region. Unfortunately, these are neither used nor mentioned although they would fill in important gaps in the study. At the moment, the study is based on local, partly incomplete and partly unpublished data sets with respect to glaciers. Some examples: (1) Regarding glacier extent, the authors always refer to the "second Chinese inventory". However, no reference to a peer-reviewed publication is provided, and the inventory only includes the Chinese parts of the headwaters, although the majority of the glacier area seems to be outside of China (Indus basin). In the Supplementary Material, the GLIMS website is found (but without a clear statement for what it was used) and without a reference to the widely-used global Randolph Glacier Inventory (Pfeffer et al., 2014). (2) Glacier mass balance data for three Chinese glaciers seem to have been used for calibration but many other mass balance measurements that are freely available from the World Glacier Monitoring Service (WGMS, 2021) are not mentioned. (3) The glacier model relies on initial glacier thickness (see Eqs in

Supplementary Material). The widely-used global data set of glacier ice thickness (Farinotti et al., 2019) is however neither mentioned nor used.

- **Set up of glacier model:** For regional to global-scale glacier modelling, about a dozen of individual models have been developed in recent years (see Marzeion et al., 2020) and have been applied to the region studied here, also to compute future glacier runoff. Not a single reference to any of these models and the respective results is made, and the approaches proposed here strongly lag behind the state-of-the-art. There is an important conceptual problem in the present model set up: Although volume-area scaling has been used in the very first version of global glaciers models, it can only (!) be applied at the scale of individual glaciers, but not for aggregates of glaciers in the hydrological response units (Representative Elementary Watersheds) employed by the authors. Due to the non-linearity of the equation, this will lead to erroneous results (too high ice thickness for an aggregate of glaciers, and thus too slow climate change response). Furthermore, the important effect of glacier retreat into higher elevations (and therefore stabilization) is neglected by the present approach.

- **Glacier model calibration:** The authors consider it as a main advance of the present study over previous research that individual water balance components have been specifically calibrated – with which I fully agree. However, for the glacier model, I can not see any actual calibration or validation. Suppl. Fig. 13 is highly misleading and does not provide any insights into model performance to compute glacier runoff: (1) The concept is difficult to understand: Trying to reproduce glacier area in sub-basins is not a validation of the water yield from glaciers. In my opinion, there even is no need for this analysis, as the glacier extent is known and therefore does not need to be modelled. (2) The high correlation coefficients are fully artificial: When plotting the area of large and small hydrological response units together this conceals relative errors for individual sub-basins.

- **Probably unrealistic water balance components (!):** Related to the above, I have tried to get more insights whether the water balance components are actually realistically modelled. Unfortunately, none of the figures and tables included in the paper and the Supplementary provides a direct indication on this and I did my own computations based on Supplementary Table 1 (providing glacier area, and mean basin runoff) and Figure 2 (providing glacier runoff contribution). From this, I computed specific annual water yield from glaciers as modelled by the authors. This can then be validated against available large-scale data sets on present glacier imbalance (Hugonnet et al., 2021) – an approach the authors should also have employed in my opinion. It comes out that glaciers in the Indus basin show specific annual water yields of more than 3 meters. This 5-10 times (!) more than indicated by the observations, and will importantly bias the authors' conclusions regarding glacier contribution to runoff. A further, and more detailed validation was not possible to me as it was unclearly defined in the manuscript what the authors actually considered as "glacier runoff" (an important omission): If it was just ice melt (after disappearance of seasonal snow) the overestimate would even be more severe. If snow melt and rain over the glacier was included in their calculations (which I do not think based on my reading), the overestimate would be a bit less important but as average specific snow melt in the whole upper Indus basin only seems to account for <0.1 m it cannot explain the misfit. These are absolutely important problems in the present set up of the model study that put the main results and conclusions into question. It might be that my assessment is not fully correct, just because I did not have access to the complete information. However, none of this was discussed or investigated by the authors, although the required data sets would be ready.

- **Definition of warming levels:** The set up of the study importantly relies on the definition of warming levels. I was surprised to see that these warming levels are not taken from GCM simulation for one consistent time period (e.g. GCMs delivering +2 deg C by the end of the century as would be intuitive) but for every single GCM just after reaching a certain threshold. From a climatological perspective, I am not sure if this

approach is admissible: Long-term responses in the climate system (e.g. for rainfall and glacier evolution) are fully ignored by this approach, and in my opinion, a GCM result with +2 deg C in 2030 or in 2070 is not directly comparable. I am however unable to judge how relevant this problem is for the final results.

- Interpretation of water scarcity: The water scarcity index links runoff changes to downstream population and is an attempt to extend the study's reach. When it comes to future changes as projected in this study, I think that there's a conceptual problem: The authors simulate changes in streamflow, i.e. a linear element in a large-scale basin. A change in streamflow will have a local effect on water availability but regions away from the main stream will be affected differently, e.g. they will depend on changes in local rainfall and ET but not changes in glacier melt. Therefore, I have the impression that a straight-forward applicability leads to questionable results that would need to be further investigated. Again, however, I might not be fully able to judge what was done because the description (Supplementary) is partly unclear.

Specific comments:

- Line 31: Here and throughout the paper: Changes in runoff between two periods are always referred to as a "rate". This is incorrect. A rate would be the change "per year". A more careful writing is required.
- Line 31: Already the abstract should at least provide some indication where the results come from (hydrological modelling)
- Line 55: This is much too short as a description of the approach used here (even though more details are given in the Methods). The authors should give at least some insights into which methods or approaches have been used. Otherwise, the results presented in the main text are difficult to be understood by the reader.
- Line 66: How was the bias-correction performed? It is clear that the main text cannot give full details, but an indication and a reference to the Methods section is required.
- Line 105: Is it really the first time that "non-monotonic" changes in river flow have been detected for different climate scenarios? I have the impression that similar results have been found in previous literature already, but I might have misunderstood the way the authors define "non-monotonic".
- Line 130: Unclear to which period of the year the decrease refers to – the annual period, or just the summer? This is just an example for the many instances where the presentation of the results is not fully complete thus hampering the interpretation.
- Line 149: The conclusion drawn here that the low-emission scenario leading to smaller temperature change (and thus smaller precipitation change and consequently slightly lower runoff) "poses a severe threat" is highly problematic and should be reformulated. Yes, high-emission scenarios lead to somewhat higher runoff at the annual scale (due to more rain) but also cause more extreme events (heat, rain) and the threat to e.g. Bangladesh in terms of sea-level rise will by far prevail...
- Line 198: Nowhere in the paper (also not Supplementary), an indication on how large these "well-defined spatial domains", representing the model resolution are. This is an essential information in my opinion.
- Line 203: This statement calls for a reference.
- Line 206/207: As it stands now, this is simply not understandable without further information. It should either be removed or be expanded.
- Line 2017: several references to the datasets missing. The validation with glacier mass balance data is not shown.
- Line 236: I would rather expect to see the results of the present study here that should be made available, and not a data availability statement for the data sets used as input for the modelling.
- Line 251: Similar comment here. Code availability is given for one small sub-package used but what about the main code and model that the study relies on?
- Figure 2: Here and elsewhere it would make sense to write out the basin names instead of using the acronyms – it would become easier accessible. See main comment above on the unrealistic (at least according to my checks) magnitude of glacier contribution (both in Indus and Brahmaputra basin).
- Supplementary Material line 21 etc: the description of the glacier module does not

contain a single glaciological reference – although there would be innumerable studies to base the chosen approaches upon.

- Supplementary Material line 36: How large is the REW? What is the elevation band spacing? Throughout the methods description many aspects remain superficial, not allowing reproducibility or understanding of the implications on the results.
- Supplementary Material line 51: The “turnover” from snow to ice is a noteworthy term that I have never heard in this context. This should be better described and introduced as I think it would be difficult to be directly relate to glaciological processes. In fact, it would be much easier (as in other glacio-hydrological models) to model the disappearance of the snow layer on top of the glacier ice and then use the degree-day factor for ice.
- Supplementary Material line 62: How were the degree-day factors calibrated? This is another example, where essential details in the model implementation are missing which makes it difficult to assess the quality of the modelling and the results.
- Supplementary Material line 71: T and P extracted from what? I assume from a data set (that is not named however).
- Supplementary Material line 99: performance for which variable?
- Supplementary Material line 106: I do not understand this statement – glacier area is known from the inventories and does not need to be calibrated. Furthermore, how is this calibrated here?
- Supplementary Material line 126: It is unclear what has actually been done to apply Eq XX (no number provided). How is Q, the total water availability, computed? If it is taken from the distributed hydrological model this is fine but if it is just catchment runoff there is a conceptual problem.

References

- Farinotti, D., Huss, M., Fürst, J. J., Landmann, J., Machguth, H., Maussion, F., & Pandit, A. (2019). A consensus estimate for the ice thickness distribution of all glaciers on Earth. *Nature Geoscience*, 12(3), 168-173.
- Hugonnet, R., McNabb, R., Berthier, E., Menounos, B., Nuth, C., Girod, L., ... & Kääb, A. (2021). Accelerated global glacier mass loss in the early twenty-first century. *Nature*, 592(7856), 726-731.
- Marzeion, B., Hock, R., Anderson, B., Bliss, A., Champollion, N., Fujita, K., ... & Zekollari, H. (2020). Partitioning the uncertainty of ensemble projections of global glacier mass change. *Earth's Future*, 8(7), e2019EF001470.
- Pfeffer, W. T., Arendt, A. A., Bliss, A., Bolch, T., Cogley, J. G., Gardner, A. S., ... & Randolph Consortium. (2014). The Randolph Glacier Inventory: a globally complete inventory of glaciers. *Journal of glaciology*, 60(221), 537-552.
- WGMS (2021): Fluctuations of Glaciers Database. World Glacier Monitoring Service, Zurich, Switzerland. DOI:10.5904/wgms-fog-2021-05. Online access: <http://dx.doi.org/10.5904/wgms-fog-2021-05>

Reviewer #3 (Remarks to the Author):

I read this paper with interest because of its implications for water security for almost a billion people living in South and South East Asia (and China).

The work reported here depends on model simulations: (1) predictions of climate (precipitation and temperature) by a suite of climate models, including bias correction, and (2) converting the predicted climate to streamflow time series with the use of a calibrated, physically based, distributed model (THREW) that includes algorithms for snow and glacier melt in cold & mountainous regions. The use of THREW is a particular strength of the methodology used (in relation to previous published work in this region).

The main outcome of the modeling study, for scientists and lay people living in this region, is that there is considerable nonuniformity in the regional streamflow response to global warming, and this nonuniformity is nonlinearly dependent upon the global warming scenarios. This is quite contrary to what has been reported before (as per the

literature review cited in the paper). This is somewhat surprising, but also intriguing (and concerning) to people living in the region. It can also be confusing, because this is not a simple story that can be communicated easily.

I am very supportive of this work, based on my confidence on the authorship of the paper and the methods/models used, including THREW (which I am very familiar with).

However, I do have some concerns about how the story is communicated to both scientists and to lay people. I believe there is more of an opportunity for the authors present it in a way that generates more confidence. Instead of stating the outcome of the model predictions as fact, the authors can do a much better job explaining and interpreting the results in a way that can be understood by the readers. Especially I would like them to explain in a process based way WHY they are getting the results they are getting. This should be straightforward, given the physically based model, THREW. For example why is the Indus River behaving differently from the others? Is it connected to regional differences in atmospheric and surface temperatures and topographic elevation that has an impact on the relative snow fraction in precipitation, and the magnitude and timing of snow and glacier melt? Readers will believe the results if they are articulated in a process based way, rather than treating internal model workings as a black box.

This richness of detail of the internal workings of the model, when added to the model predictions, will make the paper much more interesting, but also raise a lot of interest on climate change impacts on the hydrology of the Tibetan Plateau region.

I therefore recommend revisions to address these concerns before you decide to accept the paper for publication in Nature Communications.

Response to Reviewer 1

(1) The Asian water tower is one of the most sensitive areas of climate change in the world and is also the source of many great rivers. The change of water resources in its upstream will affect the living of its downstream residents. In the paper, based on the observation-constrained hydrological model, the Tsinghua Representative Elementary Watershed (THREW) model, non-monotonic changes of river flows for seven river basins, under the warming levels of 1.5, 2.0, and 3.0 °C, are found. Firstly, the characteristics of streamflow under different warming levels were described respectively. Then, the contributions of different components were analyzed. On this basis, the influence on downstream is further discussed. This study has implications for water management in Asian water towers in a warming world. My overall comments for the current paper are as follows:

Response: We thank the reviewer for the appreciation of our manuscript. We address each of your comments below on a point-to-point basis.

(2) Abstract: It would be better to specify the value of change in Indus.

Response: We specify the magnitudes of changes (i.e., both increase and decrease) for the seven river basins in the abstract.

(3) Line 57, “different global warming levels (i.e., 1.5 °C, 2.0 °C and 3 °C)” how to select the warming levels? Any reasons?

Response: The warming levels of 1.5 °C and 2.0 °C are the central aims of the Paris Agreement “to strengthen the global response to the threat of climate change by keeping a global temperature rise this century well below 2 °C above pre-industrial levels and to pursue efforts to limit the temperature increase even further to 1.5 °C” (Schleussner, 2016). We consider another warming level, i.e., 3 °C. This is the projected warming level by the end of the 21st century, given the current nationally-determined mitigation ambitions (IPCC, 2018). We made this clear in the revised manuscript (Lines 238-244).

(4) Line 58, “This is mainly associated with complex runoff components in TP river basins that respond differently as climate warms”. The sentence seems to make no sense here. I noticed a similar sentence in line 109, which implies that contributions of different runoff components are about to be analyzed.

Response: This sentence is deleted from the revised manuscript.

(5) Line 66, “from 22 Coupled Model Intercomparison”, why 22 CMIP6 models are selected? How to deal with the resolution with different CMIP6 models?

Response: The 22 CMIP6 models are selected based on previous evaluation of the models’ ability in reproducing historical temperature and precipitation over TP (see e.g., Cui et al. (2021) for details). We re-

gridded the 22 CMIP6 model output from various spatial resolutions to the resolution of observations (i.e., the WFD $0.5^\circ \times 0.5^\circ$) based on bilinear interpolation. The original temporal resolution for all the 22 CMIP6 models is daily scale, consistent with WFD. We provide details in the revised manuscript (Lines 218-224).

(6) Results: We usually consider the summer season as June-August. What is the author's consideration in identifying summer as May-July here? Therefore, the period of autumn has changed too.

Response: Thanks for pointing this out. We adopt the conventional definition of boreal summer (i.e., June to August) and autumn (September to November). We made changes accordingly in the revised manuscript. Thanks!

(7) Line 85, “The declined streamflow in summer is accompanied by significant increases in autumn streamflow (i.e., August to October) in the two river basins. The sharp contrasts in summer and autumn streamflow correspond to a shift of peak monthly streamflow from July to August by 2070s for the upper Salween and Mekong”. The phenomenon of “significant increases in autumn” is not only found in the two basins, but also in other basins, such as UYE, UYA and so on in supplementary fig.1, or even more significant. Besides, the “peak monthly streamflow” is obscure to me because I didn't draw the conclusion from the mentioned figs.

Response: We mean that the occurrence of peak streamflow shows divergent temporal shifts over TP, with delayed occurrence (i.e., from July to August) for the upper Salween and Mekong. The other basins, e.g., Yellow (UYE) and Yangtze (UYA), show increases in both summer and autumn streamflow, but the occurrence of peak streamflow does not change. We made this clear in the revised manuscript (Line 89-94).

(8) Line 89, “... over Salween and Mekong further contributes to negative ...”. I would suggest deleting the “further” here as it seems to confuse the relationship of warming at 2°C and 3°C .

Response: Revised as suggested.

(9) Line 101, “... across TP river basins except for the upper Indus”. I would suggest stating the value of Indus.

Response: We made this clear in the revised manuscript.

(10) Line 113, “There are three runoff components over TP, i.e., rainfall runoff (including rain-on-snow processes), snowmelt runoff, and glacier-melt runoff”. I would like to know why there are three components contributing to streamflow. And evapotranspiration is mentioned at the end of the paragraph. Is there any connection between them?

Response: We classify runoff components into three categories according to the hydrological processes that generate them, i.e., rainfall (including rain-on-snow/ice), snowmelt, and glacier melt. We discern snowmelt

and glacier-melt runoff due to the distinct temporal scales between the two processes in runoff generation as well as how they are represented in THREW. All three components are observed and mentioned in previous studies (e.g., Lutz et al., 2014; Pritchard et al., 2019; Yao et al., 2019), and are properly modeled in the THREW model (see *Method* for details). We mention evapotranspiration due to its role in regulating basin-scale water balance, i.e., from a sufficient long-term perspective, precipitation minus evapotranspiration (P-E) equals runoff (R).

(11) Line 124, “For the warming levels of 1.5 °C and 2.0 °C, the decreases in snowmelt runoff outweigh increased rainfall runoff especially for the upper Salween and Mekong (Fig. 3)”. The color bar in fig. 3 does not seem to agree with what the author says for the Salween. And there is an extra UM in fig. 3.

Response: Thanks for pointing out the careless mistake. The correct figure (i.e. Fig. 3) is provided in the revised manuscript. Thanks!

(12) Line 141, “... consistent increases with temperature, ranging from 3.5% to 7.3%, ...”. Does the range correspond to the change in temperature from 1.5 °C to 3 °C? Please specify it.

Response: Correct. We made this clear in the revised manuscript.

(13) Implications: Line 157, “... and peak monthly streamflow (e.g., Fig. 2 for Mekong river basin) ...”. As mentioned in 4, I do not understand how to understand the peak here.

Response: We change “peak monthly streamflow” to “the occurrence of peak streamflow” in the revised manuscript. This mostly refers to the timing. Thanks!

Response to Reviewer 2

(1) This paper describes a comprehensive hydrological modelling study to assess the response of large streams originating from the Tibetan Plateau to three different atmospheric warming levels. The authors have assembled a large amount of data sets and model approaches to provide streamflow projections and analyse the changes in water runoff from the considered headwater catchments from individual components (rain, snow melt, glacier melt). Particular attention is devoted to the calibration and validation of the components of the hydrological response to climate change and a complex pattern of runoff shifts is found, depending on the amount of atmospheric warming and the characteristics of the basin (e.g. percentage of glacier cover). Also, the effect of future changes in streamflow on water scarcity on the population in the regions surrounding the Tibetan Plateau is assessed.

Response: We thank the reviewer for the overall appreciation of our manuscript.

(2) Although it is obvious that the effort for conducting the study and setting up this comprehensive model framework was tremendous and the results are potentially interesting, I have detected a number of important weaknesses and problems in the present version of the study that – in my opinion – preclude publication in Nature Communications at the moment. These major issues are summarized below, followed by some more specific remarks. My review puts an emphasis on the representation of glaciers and their hydrological contribution.

Response: We address each of the reviewer's comments below on a point-to-point basis. In particular, we substantially calibrate the module of glacier evolution in the THREW model, and further validate its performance in runoff contribution against observations from variable sources. The annual mean streamflow for Indus decrease by 2.5% and 1.9% at the warming level of 1.5 °C and 2.0 °C relative to the present-day climate, and reverse to increase by 1.5% at the warming level of 3.0 °C.

(3) Substantive comments:- General presentation: The paper is generally well written, although at several places the language would benefit from a revision by a native English speaker.

Response: Thanks. We invited Dr. Gunter Blöschl from the Vienna University of Technology to improve the language of the manuscript. Thanks!

(4) It is however most problematic that various important aspects of the methodology are only described superficially. Even when going to the Supplementary Material, many aspects of model implementation and/or decisions in setting up the modelling framework remain unclear. Model variables are unexplained or, at least, it cannot be tracked where they are taken from. Many acronyms are not defined which makes the reading more difficult. There are also loose ends: For example, in the main text a validation of model results with glacier mass balance data is promised but nowhere a result is shown – the reader is just expected to believe that the analysis was favourable.

Response: We apologize for the careless presentation. We provide comprehensive descriptions of the THREW model configuration, in terms of its structure, parameters and optimal values, and the key modules for runoff-generation processes. The details are substantially provided in the *Methods* (Lines 254-317), *Supplementary Methods* and *Supplementary Tables* section of the revised manuscript. Thanks!

(5) Referencing: Although the main text only allows a limited number of references, the choice might be revised to be broader and take into account more non-Chinese studies performed in the region. In my opinion, several relevant references are missing many data sets (Supplementary material) are present without providing a reference at all.

Response: We carry out a substantial literature review, and include references that are relevant in the revised manuscript. We would be very glad to incorporate any other references that are accidentally missing from the current reference list. In addition, we paid special attention to the references of the datasets we used. Thanks!

(6) Data sets: At several places, the motivation for use of specific data sets was difficult to follow by me as little explanation is given. For example, the study strongly relies on the WFD data set for bias correcting GCM data. Some more information is needed why this data is deemed optimal for the region, even more as the authors have not underlain the use of the data set with a publication documenting its applicability.

Response: We justify the choice of specific datasets in the revised manuscript. Some of our arguments are specified below.

The WATCH Forcing Data (WFD) is a twentieth-century meteorological forcing dataset for land surface and hydrological models (e.g., Weedon et al., 2014). It has been applied in hydrological simulation for many river basins across the world (e.g. Aich et al., 2014; Vetter et al., 2017; Huang et al., 2017; Thompson et al., 2021). Xu et al. (2019a) particularly adopted the WFD in hydrological modeling over the Brahmaputra river basin, with reasonably good performance achieved. We also compared WFD and the precipitation data from China Meteorological Administration, and found that WFD performed very well for precipitation in the study basins. In addition, WFD shows good utilities as a reference dataset for bias correcting GCM outputs (e.g., Hempel et al., 2013). We believe testing the utility of a different dataset is beyond the scope of our study. We justify our choice in the *Methods* section (Lines 207-216).

(7) On the glacier side, there are various widely accepted large-scale data sets that have been used by basically all previous recent studies in the region. Unfortunately, these are neither used nor mentioned although they would fill in important gaps in the study. At the moment, the study is based on local, partly incomplete and partly unpublished data sets with respect to glaciers. Some examples:

Response: We remedy the issues by including more widely-used datasets in our study, as great supplements to the in-situ observations we have at more fine scales. See responses #8 to #10 below.

(8) Regarding glacier extent, the authors always refer to the “second Chinese inventory”. However, no reference to a peer-reviewed publication is provided, and the inventory only includes the Chinese parts of the headwaters, although the majority of the glacier area seems to be outside of China (Indus basin). In the Supplementary Material, the GLIMS website is found (but without a clear statement for what it was used) and without a reference to the widely-used global Randolph Glacier Inventory (Pfeffer et al., 2014).

Response: For glacier extent, we adopt the First Chinese Glacier Inventory (CGI-1) (Shi et al., 2009) and Second Chinese Glacier Inventory (SCGI) (Guo et al., 2015) for the domain within China and Randolph Glacier Inventory 6.0 (RGI 6.0) for the region outside China. We provide access to both datasets and their references in the revised manuscript.

(9) Glacier mass balance data for three Chinese glaciers seem to have been used for calibration but many other mass balance measurements that are freely available from the World Glacier Monitoring Service (WGMS, 2021) are not mentioned.

Response: We thank the reviewer for raising this concern. We decide to abandon the mass balance data of the three Chinese glaciers due to their limited spatial coverage within our model domain. We re-calibrate our model based on freely available datasets (in terms of both glacier extent and thickness). We compare the glacier mass balance results from our model against WGMS and results from a different model (e.g, Hugonnet et al., 2021). As shown in *Table R1* below, the simulated tendency of glacier mass balance agrees overall with WGMS and other model results for the Indus.

Table R1 Comparison of the glacier mass balance in the Indus basin between this present study and results extracted from two different datasets.

REW	Indus			Year
	WGMS (m/yr)	Hugonnet (2021) (m/yr)	Simulated (m/yr)	
1	0.098	0.116	-0.128	2000-2008
2	-0.178	0.159	-0.185	2000-2008
3	-0.551	0.178	-0.133	2000-2008
4	-0.174	-0.084	-0.705	2000-2008
5	-0.037	-0.067	-0.212	2000-2010
6	0.35	0.02	-0.496	2000-2010
7	-0.145		-0.346	1999-2007
8	-0.554	-0.273	-1.033	2000-2011
9	-0.775	-0.401	-0.549	2000-2011
Average	-0.218	-0.044	-0.421	

(10) The glacier model relies on initial glacier thickness (see Eqs in Supplementary Material). The widely-used global data set of glacier ice thickness (Farinotti et al., 2019; Zekollari et al., 2019) is however neither mentioned nor used.

Response: In the revised manuscript, the initial glacier thickness is computed based on the area-volume scaling relationship, and is restricted by referring to Millan et al. (2022). We use Millan et al. (2022) over Farinotti et al. (2019) or Marzeion et al. (2019), because the former provides results of more glaciers.

(11) Set up of glacier model: For regional to global-scale glacier modelling, about a dozen of individual models have been developed in recent years (see Marzeion et al., 2020) and have been applied to the region studied here, also to compute future glacier runoff. Not a single reference to any of these models and the respective results is made, and the approaches proposed here strongly lag behind the state-of-the-art.

Response: We include more widely-used datasets and related studies used for reference in the revised manuscript (see the references section in the main text and *supplementary materials*). We do not claim the novelty of our approach for glacier modeling, but believe that substantial calibration and validation enable us to obtain “right results for right reasons”. The simulated changes of glacier coverage and thickness overall agree well with other datasets (see Supplementary Fig. 13 and Table 10 for details). In addition, we argue that glacier-melt runoff might be important in small sub-basins, but account for less than 4.4% for the seven river basins except Indus (14.3%). It would be less desirable to test the utility of other glacier models in the present study. We refer the readers to Marzeion et al. (2020) for a comprehensive comparison of different types of glacier models.

(12) There is an important conceptual problem in the present model setup: Although volume-area scaling has been used in the very first version of global glaciers models, it can only (!) be applied at the scale of individual glaciers, but not for aggregates of glaciers in the hydrological response units (Representative Elementary Watersheds) employed by the authors. Due to the non-linearity of the equation, this will lead to erroneous results (too high ice thickness for an aggregate of glaciers, and thus too slow climate change response). Furthermore, the important effect of glacier retreat into higher elevations (and therefore stabilization) is neglected by the present approach.

Response: We applied volume-area scaling in each GREW (not REW) to achieve dimensionality reduction. In terms of glacier retreat and accumulation, each REW is further divided into 200 m elevation bands to account for spatial variation in precipitation and temperature over the altitudinal range. We adopt several datasets (WGMS, 2021; Hugonnet et al., 2021; Millan et al., 2022) for reference to restricting the simulated ice thickness. Our simulation results show that the transformable relationship between the volume and area could be applied at the GREW scale. We clarify the details of the glacier module in the revised manuscript (Lines 262-279).

(13) Glacier model calibration: The authors consider it as a main advance of the present study over previous research that individual water balance components have been specifically calibrated – with which I fully agree. However, for the glacier model, I can not see any actual calibration or validation. Suppl. Fig. 13 is highly misleading and does not provide any insights into model performance to compute glacier runoff: (1) The concept is difficult to understand: Trying to reproduce glacier area in sub-basins is not a validation of the water yield from glaciers. In my opinion, there even is no need for this analysis, as the glacier extent is known and therefore does not need to be modelled. (2) The high correlation coefficients are fully artificial:

When plotting the area of large and small hydrological response units together this conceals relative errors for individual sub-basins.

Response: We thank the reviewer for this critique.

The calibration of the glacier module is through the comparison of the simulated glacier extent and thickness against variable datasets. It is impossible to directly validate the simulated glacier runoff due to unavailable observations. We thus validate the performance of the glacier modules through changes in glacier mass balance, particularly for the Indus where the glacier coverage is the largest. We respectfully disagree that “the glacier extent is known, and therefore does not need to be modeled”. As the reviewer can see, the glacier extents change dramatically over the seven river basins. The reduction of glacier extent is through increased glacier melt runoff. We believe it is crucial for the model to capture this change.

We validate the performance of the glacier module at the REW scale, mainly because we are interested in water balance at the sub-basin scale. We note that the computation of glacier evolution is implemented at GREW scale. The high correlation coefficients indicate that the model captures the spatial and temporal variability of glacier evolution at the basin scale. We clarify this in the revised manuscript. Thanks!

(14) Probably unrealistic water balance components (!): Related to the above, I have tried to get more insights whether the water balance components are actually realistically modelled. Unfortunately, none of the figures and tables included in the paper and the Supplementary provides a direct indication on this and I did my own computations based on Supplementary Table 1 (providing glacier area, and mean basin runoff) and Figure 2 (providing glacier runoff contribution). From this, I computed specific annual water yield from glaciers as modelled by the authors. This can then be validated against available large-scale data sets on present glacier imbalance (Hugonnet et al., 2021) – an approach the authors should also have employed in my opinion. It comes out that glaciers in the Indus basin show specific annual water yields of more than 3 meters. This 5-10 times (!) more than indicated by the observations, and will importantly bias the authors’ conclusions regarding glacier contribution to runoff. A further, and more detailed validation was not possible to me as it was unclearly defined in the manuscript what the authors actually considered as “glacier runoff” (an important omission): If it was just ice melt (after disappearance of seasonal snow) the overestimate would even be more severe. If snow melt and rain over the glacier was included in their calculations (which I do not think based on my reading), the overestimate would be a bit less important but as average specific snow melt in the whole upper Indus basin only seems to account for <0.1 m it cannot explain the misfit. These are absolutely important problems in the present set up of the model study that put the main results and conclusions into question. It might be that my assessment is not fully correct, just because I did not have access to the complete information. However, none of this was discussed or investigated by the authors, although the required data sets would be ready.

Response: We re-calibrated the model in the revised manuscript. For precipitation, most gridded datasets are of coarser resolution and insufficient quality to represent the magnitude of precipitation over complex topography of the glacierized catchments, such as the Upper Indus (e.g., Immerzeel et al., 2015). The mean annual precipitation from the WFD for the Upper Indus is 375 mm. This is much lower than 575 mm

estimated by previous studies (e.g., Shafeeque et al., 2019; Liaqat et al., 2022). Thus, we multiply the precipitation of WFD (357 mm) by 1.61 to correct the precipitation. Similar multiplication is carried out for glacier runoff simulation previously, see Lutz et al. (2014, 2016) and Immerzeel et al. (2015) for details. Rainfall runoff contributes approximately 44% of annual runoff for Indus in the revised modeling results (Fig.2).

(15) Definition of warming levels: The set up of the study importantly relies on the definition of warming levels. I was surprised to see that these warming levels are not taken from GCM simulation for one consistent time period (e.g. GCMs delivering +2 deg C by the end of the century as would be intuitive) but for every single GCM just after reaching a certain threshold. From a climatological perspective, I am not sure if this approach is admissible: Long-term responses in the climate system (e.g. for rainfall and glacier evolution) are fully ignored by this approach, and in my opinion, a GCM result with +2 deg C in 2030 or in 2070 is not directly comparable. I am however unable to judge how relevant this problem is for the final results.

Response: Due to the intrinsic distinctions of model representations, different GCMs lead to diverse climate projections at a fixed period. Inter-comparisons of hydrological responses to different model projections would thus be less desirable. Here we adopt the fixed warming levels, which is also known as the time-sampling approach as suggested in the Inter-Sectoral Impact Model Intercomparison Project (e.g., Frieler et al., 2017). This approach has already been widely accepted by the community to investigate climate impacts at different global warming levels on water resources (e.g. Gosling et al., 2017; Thompson et al., 2021), hydrological extremes (e.g. Huang et al., 2018; Marx et al., 2018; Ji et al., 2020, Klutse et al., 2018), and aquatic ecosystems (e.g. Barbarossa et al., 2021). We justify this in the revised manuscript (Lines 243-252).

(16) Interpretation of water scarcity: The water scarcity index links runoff changes to downstream population and is an attempt to extend the study's reach. When it comes to future changes as projected in this study, I think that there's a conceptual problem: The authors simulate changes in streamflow, i.e. a linear element in a large-scale basin. A change in streamflow will have a local effect on water availability but regions away from the main stream will be affected differently, e.g. they will depend on changes in local rainfall and ET but not changes in glacier melt. Therefore, I have the impression that a straight-forward applicability leads to questionable results that would need to be further investigated. Again, however, I might not be fully able to judge what was done because the description (Supplementary) is partly unclear.

Response: We provide more details about the calculation of water scarcity index in the revised manuscript (Lines 319-331). However, we respectfully disagree with the reviewer that changes in streamflow of upstream regions are not important for downstream countries. The water conflicts among the transboundary rivers are especially serious in southern Asia (Rashid et al., 2018; Basharat, 2019; Janjua et al., 2021; Orr, 2022). The upstream river flows play an important role in water demand of these downstream countries. For instance, 11% and 27% of the water resources are generated by the upstream Ganges and Indus River basin, respectively. We agree with the reviewer that this issue might be less important for the Yangtze and Yellow rivers where the tributaries and local rainfall minus ET would matter the most.

(17) Specific comments:- Line 31: Here and throughout the paper: Changes in runoff between two periods are always referred to as a “rate”. This is incorrect. A rate would be the change “per year”. A more careful writing is required.

Response: Deleted as suggested.

(18) Line 31: Already the abstract should at least provide some indication where the results come from (hydrological modelling)

Response: The abstract has been revised.

(19) Line 55: This is much too short as a description of the approach used here (even though more details are given in the Methods). The authors should give at least some insights into which methods or approaches have been used. Otherwise, the results presented in the main text are difficult to be understood by the reader.

Response: We re-construct this paragraph to provide more details. Thanks!

(20) Line 66: How was the bias-correction performed? It is clear that the main text cannot give full details, but an indication and a reference to the Methods section is required.

Response: We refer the readers to the *Methods* section for more details.

(21) Line 105: Is it really the first time that “non-monotonic” changes in river flow have been detected for different climate scenarios? I have the impression that similar results have been found in previous literature already, but I might have misunderstood the way the authors define “non-monotonic”.

Response: By “non-monotonic”, we mean annual runoff and streamflow over major TP river basins decrease with moderate temperature increases (less than 2 °C), and then revert to increase if the temperature increment is 3 °C. Our results are sharply contrasted with previous studies that show a stable or increasing tendency of runoff. For instance, Lutz et al. (2014) show that the total runoff of the five upstream river basins over TP is expected to 4.1-10.0% increase in 2041-2050. Wang et al. (2021) found 1.0-7.2% increases in runoff over major rivers of TP by the end of this century. In addition, the third anonymous reviewer also confirms that our study presents new findings in terms of runoff changes over TP. Thanks all the same!

(22) Line 130: Unclear to which period of the year the decrease refers to – the annual period, or just the summer? This is just an example for the many instances where the presentation of the results is not fully complete thus hampering the interpretation.

Response: We thank the reviewer for pointing this out. For this particular instance, we mean the annual snow coverage. We checked other instances to be more specific. Thanks!

(23) Line 149: The conclusion drawn here that the low-emission scenario leading to smaller temperature change (and thus smaller precipitation change and consequently slightly lower runoff) “poses a severe threat” is highly problematic and should be reformulated. Yes, high-emission scenarios lead to somewhat higher runoff at the annual scale (due to more rain) but also cause more extreme events (heat, rain) and the threat to e.g. Bangladesh in terms of sea-level rise will by far prevail...

Response: The reviewer is correct. Both low- and high-emission scenarios pose threats, even though they manifest in different ways. We revise the conclusions in the manuscript (Lines 188-191). Thanks!

(24) Line 198: Nowhere in the paper (also not Supplementary), an indication on how large these “well-defined spatial domains”, representing the model resolution are. This is an essential information in my opinion.

Response: We provide statistics of hydrological response units (i.e., REW) in *Table S3*. “well-defined spatial domains” is removed from the revised manuscript.

(25) Line 203: This statement calls for a reference.

Response: Done as suggested (Line292).

(26) Line 206/207: As it stands now, this is simply not understandable without further information. It should either be removed or be expanded.

Response: Removed as suggested.

(27) Line 207: several references to the datasets missing. The validation with glacier mass balance data is not shown.

Response: We add references to the datasets.

(28) Line 236: I would rather expect to see the results of the present study here that should be made available, and not a data availability statement for the data sets used as input for the modelling.

Response: Done as suggested.

(29) Line 251: Similar comment here. Code availability is given for one small sub-package used but what about the main code and model that the study relies on?

Response: We will make the main codes that the model relies on publicly available once our manuscript is conditionally accepted.

(30) Figure 2: Here and elsewhere it would make sense to write out the basin names instead of using the acronyms – it would become easier accessible. See main comment above on the unrealistic (at least according to my checks) magnitude of glacier contribution (both in Indus and Brahmaputra basin).

Response: The full basin names of each basin are added and the acronyms are removed. We re-built the model based on glacier extent and thickness datasets in Brahmaputra and Indus basins. The glacier runoff contributes 4.4% and 14.3% of total runoff for Brahmaputra and Indus basins (Fig 2).

(31) Supplementary Material line 21 etc: the description of the glacier module does not contain a single glaciological reference – although there would be innumerable studies to base the chosen approaches upon.

Response: We add glaciological references to the glacier module in the revised manuscript. Thanks!

(32) Supplementary Material line 36: How large is the REW? What is the elevation band spacing? Throughout the methods description many aspects remain superficial, not allowing reproducibility or understanding of the implications on the results.

Response: Details of the REW information are provided in *Table S3*. The elevation band spacing is 200 m. We provide this information in the revised manuscript.

(33) Supplementary Material line 51: The “turnover” from snow to ice is a noteworthy term that I have never heard in this context. This should be better described and introduced as I think it would be difficult to be directly relate to glaciological processes. In fact, it would be much easier (as in other glacio-hydrological models) to model the disappearance of the snow layer on top of the glacier ice and then use the degree-day factor for ice.

Response: By “turnover” (Luo et al., 2013), we mean ice formation (ie., the process of snow transferring to ice). We delete the term in the revised manuscript. In the study, the disappearance of the snow layer on top of the glacier ice is simulated in the THREW model by simulating this transferring process and the melting process of the snow layer on top of the glacier (see *supplementary materials* Formulas 6-9). After the top snow layer disappears (either melting or transferring to ice), the ice melting is simulated using the degree-day factor (see *supplementary materials* Formula 10). We clarify this in the revised manuscript.

(34) Supplementary Material line 62: How were the degree-day factors calibrated? This is another example, where essential details in the model implementation are missing which makes it difficult to assess the quality of the modelling and the results.

Response: We provide details of model calibration in the revised manuscript (Lines 65-71).

(35) Supplementary Material line 71: T and P extracted from what? I assume from a data set (that is not named however).

Response: T and P are extracted from the WFD and CMIP6 models. We made this clear in the revised manuscript.

(36) Supplementary Material line 99: performance for which variable?

Response: We mean the streamflow simulation. We made this clear in the revised manuscript.

(37) Supplementary Material line 106: I do not understand this statement – glacier area is known from the inventories and does not need to be calibrated. Furthermore, how is this calibrated here?

Response: In this study, the First Chinese Glacier Inventory (CGI-1) is adopted as input for model. Parameters related to glaciers (e.g. parameters in glacier volume-area scaling m_g and n_g , *Supplementary material Table 7*) is calibrated to match the simulated glacier extents and thickness with observations. The “observed” glacier extents and thickness for calibration are obtained from Second Chinese Glacier Inventory (SCGI), RGI6.0, and Millan et al (2022). See response #8, 9, 10 for details.

(38) Supplementary Material line 126: It is unclear what has actually been done to apply Eq XX (no number provided). How is Q, the total water availability, computed? If it is taken from the distributed hydrological model this is fine but if it is just catchment runoff there is a conceptual problem.

Response: Q_i was estimated using daily total runoff at 0.5° spatial resolution downloaded from ISIMIP2a. In this study, the population facing water scarcity during 1991-2010 for seven river basins is calculated based on WSI and the Gridded Population of the World collection (GPW) (*Methods*). We changed the total water availability Q_i at different warming levels based on the relative change in this study for each basin, and projected the population under water scarcity in the future. We made this clear in the revised manuscript (Lines 319-346).

Response to Reviewer 3

(1) I read this paper with interest because of its implications for water security for almost a billion people living in South and South East Asia (and China). The work reported here depends on model simulations: (a) predictions of climate (precipitation and temperature) by a suite of climate models, including bias correction, and (b) converting the predicted climate to streamflow time series with the use of a calibrated, physically based, distributed model (THREW) that includes algorithms for snow and glacier melt in cold & mountainous regions. The use of THREW is a particular strength of the methodology used (in relation to previous published work in this region). The main outcome of the modeling study, for scientists and lay people living in this region, is that there is considerable nonuniformity in the regional streamflow response to global warming, and this nonuniformity is nonlinearly dependent upon the global warming scenarios. This is quite contrary to what has been reported before (as per the literature review cited in the paper). This is somewhat surprising, but also intriguing (and concerning) to people living in the region. It can also be confusing, because this is not a simple story that can be communicated easily. I am very supportive of this work, based on my confidence on the authorship of the paper and the methods/models used, including THREW (which I am very familiar with).

Response: We thank the reviewer for the appreciation of our study.

(2) However, I do have some concerns about how the story is communicated to both scientists and to lay people. I believe there is more of an opportunity for the authors present it in a way that generates more confidence. Instead of stating the outcome of the model predictions as fact, the authors can do a much better job explaining and interpreting the results in a way that can be understood by the readers. Especially I would like them to explain in a process based way WHY they are getting the results they are getting. This should be straightforward, given the physically based model, THREW. For example why is the Indus River behaving differently from the others? Is it connected to regional differences in atmospheric and surface temperatures and topographic elevation that has an impact on the relative snow fraction in precipitation, and the magnitude and timing of snow and glacier melt? Readers will believe the results if they are articulated in a process based way, rather than treating internal model workings as a black box. This richness of detail of the internal workings of the model, when added to the model predictions, will make the paper much more interesting, but also raise a lot of interest on climate change impacts on the hydrology of the Tibetan Plateau region. I therefore recommend revisions to address these concerns before you decide to accept the paper for publication in Nature Communications.

Response: We thank the reviewer for the constructive suggestions. We provide more physical reasoning for the results we obtained from the modeling framework. For instance, we explain changes in rainfall runoff and snowmelt runoff by directly linking the results with evidence, i.e., changes in rainfall and snow coverage, to shed light on the internal workings of the model. We also explain the difference of the Indus by comparing it against other river basins. The difference is mainly associated with the less dominance of monsoonal rainfall in hydrological regimes over the Indus and the largest snow and glacier coverage that make rainfall-runoff less dominant in its streamflow. The advanced occurrence of peak flow for the Indus is also tied to earlier snow melt and glacier melt. Despite these differences, the tendency of changes in river flows, i.e.,

decrease in river flows at the warming levels of 1.5 °C and 2.0 °C and increases at the warming level of 3.0 °C, for the Indus is consistent with other river basins in the revised version. Thanks!

Reference

1. Aich, V., Liersch, S., Vetter, T., Huang, S., Tecklenburg, J., Hoffmann, P., ... & Hattermann, F. F. (2014). Comparing impacts of climate change on streamflow in four large African river basins. *Hydrology and Earth System Sciences*, 18(4), 1305-1321.
2. Barbarossa, V., Bosmans, J., Wanders, N., King, H., Bierkens, M. F., Huijbregts, M. A., & Schipper, A. M. (2021). Threats of global warming to the world's freshwater fishes. *Nature communications*, 12(1), 1-10.
3. Basharat, M. (2019). Water management in the Indus Basin in Pakistan: challenges and opportunities. *Indus River Basin*, 375-388.
4. Cui, T., Li, C., & Tian, F. (2021). Evaluation of temperature and precipitation simulations in CMIP6 models over the Tibetan Plateau. *Earth and Space Science*, 8(7), e2020EA001620.
5. Farinotti, D., Huss, M., Fürst, J. J., Landmann, J., Machguth, H., Maussion, F., & Pandit, A. (2019). A consensus estimate for the ice thickness distribution of all glaciers on Earth. *Nature Geoscience*, 12(3), 168-173.
6. Frieler, K., Lange, S., Piontek, F., Reyer, C. P., Schewe, J., Warszawski, L., ... & Yamagata, Y. (2017). Assessing the impacts of 1.5 C global warming—simulation protocol of the Inter-Sectoral Impact Model Intercomparison Project (ISIMIP2b). *Geoscientific Model Development*, 10(12), 4321-4345.
7. Gosling, S. N., Zaherpour, J., Mount, N. J., Hattermann, F. F., Dankers, R., Arheimer, B., ... & Zhang, X. (2017). A comparison of changes in river runoff from multiple global and catchment-scale hydrological models under global warming scenarios of 1 °CC, 2 °CC and 3 °CC. *Climatic Change*, 141(3), 577-595.
8. Guo, W., Liu, S., Xu, J., Wu, L., Shangguan, D., Yao, X., ... & Jiang, Z. (2015). The second Chinese glacier inventory: data, methods and results. *Journal of Glaciology*, 61(226), 357-372.
9. Hempel, S., Frieler, K., Warszawski, L., Schewe, J., & Piontek, F. (2013). A trend-preserving bias correction—the ISI-MIP approach. *Earth System Dynamics*, 4(2), 219-236.
10. Huang, S., Kumar, R., Flörke, M., Yang, T., Hundecha, Y., Kraft, P., ... & Krysanova, V. (2017). Evaluation of an ensemble of regional hydrological models in 12 large-scale river basins worldwide. *Climatic Change*, 141(3), 381-397.
11. Huang, S., Kumar, R., Rakovec, O., Aich, V., Wang, X., Samaniego, L., ... & Krysanova, V. (2018). Multimodel assessment of flood characteristics in four large river basins at global warming of 1.5, 2.0 and 3.0 K above the pre-industrial level. *Environmental Research Letters*, 13(12), 124005.
12. Hugonnet, R., McNabb, R., Berthier, E., Menounos, B., Nuth, C., Girod, L., ... & Kääb, A. (2021). Accelerated global glacier mass loss in the early twenty-first century. *Nature*, 592(7856), 726-731.
13. Immerzeel, W. W., Wanders, N., Lutz, A. F., Shea, J. M., & Bierkens, M. F. P. (2015). Reconciling high-altitude precipitation in the upper Indus basin with glacier mass balances and runoff. *Hydrology and Earth System Sciences*, 19(11), 4673-4687.
14. IPCC (2018). Global warming of 1.5 C :An IPCC Special Report on the impacts of global warming of 1.5 C above pre-industrial levels and related global greenhouse gas emission pathways, in the context

- of strengthening the global response to the threat of climate change, sustainable development, and efforts to eradicate poverty. World Meteorological Organization, Geneva, Switzerland, 32 pp.
15. Janjua, S., Hassan, I., Muhammad, S., Ahmed, S., & Ahmed, A. (2021). Water management in Pakistan's Indus Basin: challenges and opportunities. *Water Policy*, 23(6), 1329-1343.
 16. Ji, P., Yuan, X., Ma, F., & Pan, M. (2020). Accelerated hydrological cycle over the Sanjiangyuan region induces more streamflow extremes at different global warming levels. *Hydrology and Earth System Sciences*, 24(11), 5439-5451.
 17. Klutse, N. A. B., Ajayi, V. O., Gbobaniyi, E. O., Egbebiyi, T. S., Kouadio, K., Nkrumah, F., ... & Dosio, A. (2018). Potential impact of 1.5 °CC and 2 °CC global warming on consecutive dry and wet days over West Africa. *Environmental Research Letters*, 13(5), 055013.
 18. Liaqat, M. U., Grossi, G., & Ranzi, R. (2022). Characterization of interannual and seasonal variability of hydro-climatic trends in the Upper Indus Basin. *Theoretical and Applied Climatology*, 147(3), 1163-1184.
 19. Lutz, A. F., Immerzeel, W. W., Kraaijenbrink, P. D., Shrestha, A. B., & Bierkens, M. F. (2016). Climate change impacts on the upper Indus hydrology: sources, shifts and extremes. *PloS one*, 11(11), e0165630.
 20. Lutz, A. F., Immerzeel, W. W., Shrestha, A. B., & Bierkens, M. F. P. (2014). Consistent increase in High Asia's runoff due to increasing glacier melt and precipitation. *Nature Climate Change*, 4(7), 587-592.
 21. Lutz, A. F., Immerzeel, W., & Kraaijenbrink, P. (2014). Gridded meteorological datasets and hydrological modelling in the upper Indus Basin. Final Report, for International Centre for Integrated Mountain Development (ICIMOD), FutureWater, Costerweg, 1, 6702.
 22. Marx, A., Kumar, R., Thober, S., Rakovec, O., Wanders, N., Zink, M., ... & Samaniego, L. (2018). Climate change alters low flows in Europe under global warming of 1.5, 2, and 3 °C. *Hydrology and Earth System Sciences*, 22(2), 1017-1032.
 23. Marzeion, B., Hock, R., Anderson, B., Bliss, A., Champollion, N., Fujita, K., ... & Zekollari, H. (2020). Partitioning the uncertainty of ensemble projections of global glacier mass change. *Earth's Future*, 8(7), e2019EF001470.
 24. Millan, R., Mouginot, J., Rabatel, A., & Morlighem, M. (2022). Ice velocity and thickness of the world's glaciers. *Nature Geoscience*, 15(2), 124-129.
 25. Orr, A., Ahmad, B., Alam, U., Appadurai, A., Bharucha, Z. P., Biemans, H., ... & Wescoat Jr, J. L. (2022). Knowledge priorities on climate change and water in the Upper Indus Basin: A horizon scanning exercise to identify the top 100 research questions in social and natural sciences. *Earth's Future*, 10(4), e2021EF002619.
 26. Pritchard, H. D. (2019). Asia's shrinking glaciers protect large populations from drought stress. *Nature*, 569(7758), 649-654.
 27. Rashid, M. U., Latif, A., & Azmat, M. (2018). Optimizing irrigation deficit of multipurpose cascade reservoirs. *Water resources management*, 32(5), 1675-1687.
 28. RGI Consortium .(2017). Randolph Glacier Inventory – A Dataset of Global Glacier Outlines: Version 6.0: Technical Report, Global Land Ice Measurements from Space, Colorado, USA. Digital Media. DOI: <https://doi.org/10.7265/N5-RGI-60>.

29. Schleussner, C. F., Rogelj, J., Schaeffer, M., Lissner, T., Licker, R., Fischer, E. M., ... & Hare, W. (2016). Science and policy characteristics of the Paris Agreement temperature goal. *Nature Climate Change*, 6(9), 827-835.
30. Shafeeque, M., Luo, Y., Wang, X., & Sun, L. (2019). Revealing vertical distribution of precipitation in the glacierized upper Indus basin based on multiple datasets. *Journal of Hydrometeorology*, 20(12), 2291-2314.
31. Shi, Y., Liu, C., & Kang, E. (2009). The glacier inventory of China. *Annals of Glaciology*, 50(53), 1-4.
32. Thompson, J. R., Gosling, S. N., Zaherpour, J., & Laizé, C. L. R. (2021). Increasing risk of ecological change to major rivers of the world with global warming. *Earth's Future*, 9(11), e2021EF002048.
33. Vetter, T., Huang, S. H., Aich, V., Yang, T., Wang, X., Krysanova, V., & Hattermann, F. (2015). Multi-model climate impact assessment and intercomparison for three large-scale river basins on three continents. *Earth System Dynamics*, 6(1), 17-43.
34. Wang, T., Zhao, Y., Xu, C., Ciais, P., Liu, D., Yang, H., ... & Yao, T. (2021). Atmospheric dynamic constraints on Tibetan Plateau freshwater under Paris climate targets. *Nature Climate Change*, 11(3), 219-225.
35. Weedon, G. P., Balsamo, G., Bellouin, N., Gomes, S., Best, M. J., & Viterbo, P. (2014). The WFDEI meteorological forcing data set: WATCH Forcing Data methodology applied to ERA-Interim reanalysis data. *Water Resources Research*, 50(9), 7505-7514.
36. WGMS: Fluctuations of Glaciers Database. World Glacier Monitoring Service, Zurich, Switzerland. (2021). DOI:10.5904/wgms-fog-2021-05. Online access: <http://dx.doi.org/10.5904/wgms-fog-2021-05>.
37. Xu, R., Hu, H., Tian, F., Li, C., & Khan, M. Y. A. (2019). Projected climate change impacts on future streamflow of the Yarlung Tsangpo-Brahmaputra River. *Global and Planetary Change*, 175, 144-159.
38. Yao, T., Xue, Y., Chen, D., Chen, F., Thompson, L., Cui, P., ... & Li, Q. (2019). Recent third pole's rapid warming accompanies cryospheric melt and water cycle intensification and interactions between monsoon and environment: Multidisciplinary approach with observations, modeling, and analysis. *Bulletin of the American Meteorological Society*, 100(3), 423-444.

Reviewer #1 (Remarks to the Author):

My comment has been replied.

Reviewer #2 (Remarks to the Author):

The authors have provided a thorough revision of their first submission in response to three reviews. Whereas several important aspects have been improved, e.g. regarding a somewhat more detailed description of the methodology, some quite relevant aspects remain unclear and the issues raised in my first review have not (or not fully) been answered. Therefore, I still see a considerable need for further work to make this paper acceptable for publication in Nature Communications. I am mainly judging on the glaciological aspect of the study. Of course, this just represents one part of the comprehensive model framework that is used in this study but as most non-linear changes (i.e. a focus of this work) stem from transient glacier changes, I consider this part as highly important in the present context.

Below, I just outline a few issues that require further attention and were not sufficiently or inappropriately addressed in the revision:

- **Data sets: Some changes are too superficial. For example, I raised the concern that the volume-area scaling approach to estimate glacier volume used by the authors is outdated since more than a decade and that widely accepted and new data sets (e.g. Millan et al., 2022, Nature Geoscience) are available today, covering all glaciers of the study region. In response to that comment, the authors just added a reference to a new study stating that their result is more or less the same (without a proof) and continue using their previous data set. For a major revision, I would have expected more effort.**
- **In my first review, I also had major concerns regarding the validation of the model regarding the rate of glacier storage change, i.e. the main driver for non-linear changes in catchment runoff. The authors have now included a table with measured glacier mass balances of the last two decades based on Hugonnet et al. (2021), Nature, and conclude that their results are in good agreement. (As a side note: The authors termed those results "modelled" although they are based on remotely-sensed observations, i.e. measured.) Indeed, on average, the rates of mass loss seem to be (visually) matched but no numbers or statistics are provided. This is not acceptable for a scientific publication in my opinion. Furthermore, there seems to be a complete lack of correlation between measured and modelled regional mass loss rates, which is however not even mentioned. The authors' evaluation thus appears too optimistic and fully uncritical.**
- **Related to the above, I did not see my important point of the first review on the rates of glacier mass loss recomputed on my own based on Supplementary Table 1 (providing glacier area, and mean basin runoff) and Figure 2 (providing glacier runoff contribution) answered: the authors did not comment on this concern and also did not try to disentangle my problems in assessing their results. It might be that my approach of recomputing glacier mass changes based on the results presented in Figure 2 is biased as the authors still do not define what they actually consider as "glacier runoff" (still an important omission).**

In summary, I am convinced that further work is needed. Also, the presentation of the final conclusions regarding water scarcity for different warming levels is delicate in my opinion as the definition of warming levels still is difficult to be followed from a climatological point of view (not tied to a single point in time). Furthermore, the interpretation can be misleading as higher warming can result in smaller water scarcity and therefore might even seem favourable (because all other negative effects on the climate system with high warming are not focussed on...). This latter statement does not question the publication's results and findings in general but the way the results are conveyed to a broader public.

Response to Reviewer 1

My comment has been replied.

Response: We really appreciate your previous comments that greatly help us to improve the manuscripts. Thanks!

Response to Reviewer 2

(1) Data sets: Some changes are too superficial. For example, I raised the concern that the volume-area scaling approach to estimate glacier volume used by the authors is outdated since more than a decade and that widely accepted and new data sets (e.g. Millan et al., 2022, Nature Geoscience) are available today, covering all glaciers of the study region. In response to that comment, the authors just added a reference to a new study stating that their result is more or less the same (without a proof) and continue using their previous data set. For a major revision, I would have expected more effort.

Response: We thank the reviewer for once again raising this issue. We respectfully disagree that the volume-scaling approach is out-of-date just because it is mathematically simple compared to other approaches (e.g., grid-based models). To the best of our knowledge, the volume-scaling approach is still being widely used for glacier simulation in Asia's high mountains (Lutz et al., 2016; Sakai & Fujita, 2017; Banerjee & Jadhav, 2020; Shi et al., 2020; Gopika et al., 2021; Zhang et al., 2022). Previous studies also show that the volume-scaling approach is close to physical models in simulating glacier changes (Marzeion et al., 2020). Alternative sophisticated approaches rely on fully coupled ice models, and require explicit observations and reconstruction records from glaciological and geodetic methods. Those approaches might be more desired for fine-scale simulation at the glacier scale, but would not be feasible for watersheds that consist of hundreds of glaciers.

We agree that uncertainty remains using the volume-scaling approach. However, we assure that the uncertainty in terms of glacier simulation would less likely jeopardize our main conclusion, i.e., the non-monotonic changes in river flow under different warming scenarios. This is mainly because glacier runoff contributes to less than 15% of the total runoff for our basins. To further justify our argument, we carry out sensitivity experiments that specially target the component of glacier simulation.

In our last revision, we did apply the new glacier datasets by following the reviewer's suggestion in our previous revision. The reviewer may accidentally ignore our revision. Actually, the initial glacier thickness was set based on adjusted Randolph Glacier Inventory 6.0 (RGI6.0) and Millan et al. (2022) by restricting pertinent parameters (m_g , n_g) (Equation 11 and Equation 12 in the Supplementary). Since the accurate change rates of ice thicknesses and mass balance are unknown, a range of modelled ice thicknesses (0 ~ 400m) and a mean mass-balance rate (0.4 m/yr) are adopted during the calibration procedure (WGMS; Hugonnet et al., 2021). In this revision, we carried out sensitivity experiments that target the parameters used in the volume-area scaling approach, to highlight that our conclusion on the non-monotonic changes of river flows is still valid (Supplementary Method, also see our response to editor if it is available to reviewer). We provided the details of our sensitivity experiments in this revised Supplementary Method. Thanks!

(2) The authors have now included a table with measured glacier mass balances of the last two decades based on Hugonnet et al. (2021), Nature, and conclude that their results are in good agreement. (As a side note: The

authors termed those results “modelled” although they are based on remotely-sensed observations, i.e. measured.) Indeed, on average, the rates of mass loss seem to be (visually) matched but no numbers or statistics are provided. This is not acceptable for a scientific publication in my opinion. Furthermore, there seems to be a complete lack of correlation between measured and modelled regional mass loss rates, which is however not even mentioned. The authors’ evaluation thus appears too optimistic and fully uncritical.

Response: Thanks for pointing this out. We added the correlation coefficients of glacier mass balance in this revised manuscript. The correlation coefficient for the tendency of glacier mass balance between our simulation and Hugonnet et al. (2021) is 0.73 (P=0.05). Our simulation also is consistent with WGMS, with a correlation coefficient of 0.31. The mean change rate of all sub-watersheds for the Indus is -0.42 m/yr from the simulation. The mean change rate is relatively larger than either WGMS (i.e., -0.22 m/yr) or Hugonnet et al. (2021) (i.e., -0.04 m/yr). This is expected as we notice that the two remotely-based observations demonstrate variations themselves in either the basin-wide change rates or over each sub-watershed. The presented simulation bias can be either related to the uncertainty in remotely-based observations or our model simulation (particularly the glacier module). We carried out sensitivity analyses for two important glacier module parameters, i.e., the annual glacier changes rate and degree-day factors, over the Indus (Supplementary Method). Results show the changes in parameters do not influence our conclusion in terms of non-monotonic changes in total runoff at different warming levels (Supplementary Table 11). This indicates that the glacier probably plays a minor role in the uncertainty. We changed the expression of "modelled" to "remote-sensing based" in our revised manuscript by following the reviewer's suggestion. Thanks!

(3) Related to the above, I did not see my important point of the first review on the rates of glacier mass loss recomputed on my own based on Supplementary Table 1 (providing glacier area, and mean basin runoff) and Figure 2 (providing glacier runoff contribution) answered: the authors did not comment on this concern and also did not try to disentangle my problems in assessing their results. It might be that my approach of recomputing glacier mass changes based on the results presented in Figure 2 is biased as the authors still do not define what they actually consider as “glacier runoff” (still an important omission).

Response: In our study, the liquid form of precipitation over the glacier is regarded as rainfall runoff. The glacier runoff is defined as runoff contributed by glacier melt or snowmelt over the glacier coverage. We made this clear in the revised manuscript (Lines 290-293). We understand the reviewer’s approach to computing glacier mass change. In our previous version, the glacier runoff is $471 \text{ mm} \times 51\% \approx 235 \text{ mm}$. Considering the spatial extent of the glacier (i.e. 12335 km^2 glacier area based on the previous dataset), which is 7.4 % of the drainage area, this needs at least 3.1 m glacier annual melt to produce 235 mm glacier runoff. After examining our hydrological models, we found that basin-scale precipitation has been considerably underestimated. We carry out bias-correction for the annual mean basin-average precipitation (from 375 mm to 575 mm, similarly also see Lutz et al., 2014,2016; Immerzeel et al., 2015; Shafeeque et al, 2019; Liaqat et al., 2022 for details), the percentage of glacier runoff for the Indus is 14 %. This gives us 65.9 mm glacier runoff and around 0.57 m glacier and snow melt over the glaciers (i.e., 19058 km^2 glacier area based on RGI). Thanks!

(4) the presentation of the final conclusions regarding water scarcity for different warming levels is delicate in my opinion as the definition of warming levels still is difficult to be followed from a climatological point of view (not tied to a single point in time).

Response: We thank the reviewer for raising this concern. But it is not feasible to tie different warming levels to a single point of time. This is associated with the intrinsic uncertainty of the climate system and the contrasting representations of the uncertainty in CMIP6 models. We determine the warming period if the 30-year running mean global surface temperature for the first time exceeds the targeted warming level. This method is known as the time-sampling method. It is recommended by the Inter-Sectoral Impact Model Intercomparison Project (ISIMIP). The time-sampling method has already been widely accepted by the community to investigate climate impacts at different global warming levels on water scarcity (e.g. Schewe et al., 2014; Koutroulis et al., 2019), and water resources (e.g. Gosling et al., 2017; Thompson et al., 2021). Based on these facts, we prefer not to change the corresponding texts in the manuscript. Thanks all the same!

(5) Furthermore, the interpretation can be misleading as higher warming can result in smaller water scarcity and therefore might even seem favourable (because all other negative effects on the climate system with high warming are not focussed on...). This latter statement does not question the publication's results and findings in general but the way the results are conveyed to a broader public.

Response: We thank the reviewer for pointing this out. Yes, our major conclusion of non-monotonic changes in river flows does imply that higher warming leads to more precipitation and the domination of rainfall-runoff for river flows and thus less water scarcity for the study area. We believe that our results will update our existing knowledge of the hydrological responses to climate change over the Tibetan Plateau, and are thus critical for developing effective strategies for water resources management. We acknowledge the reviewer's point of adverse impacts associated with climate warming. We explicitly mention them in the revised manuscript, which include more frequent flooding and changes in flood peak timing (see Lines 195-204). Thanks!

References

- Banerjee, A., Patil, D., & Jadhav, A. (2020). Possible biases in scaling-based estimates of glacier change: a case study in the Himalaya. *The Cryosphere*, 14(9), 3235-3247.
- Gopika, J. S., Kulkarni, A. V., Prasad, V., Srinivasalu, P., & Raman, A. (2021). Estimation of glacier stored water in the Bhaga basin using laminar flow and volume-area scaling methods. *Remote Sensing Applications: Society and Environment*, 24, 100656.
- Gosling, S. N., Zaherpour, J., Mount, N. J., Hattermann, F. F., Dankers, R., Arheimer, B., ... & Zhang, X. (2017). A comparison of changes in river runoff from multiple global and catchment-scale hydrological models under global warming scenarios of 1 °C, 2 °C and 3 °C. *Climatic Change*, 141(3), 577-595.
- Hugonnet, R., McNabb, R., Berthier, E., Menounos, B., Nuth, C., Girod, L., ... & Kääb, A. (2021). Accelerated global glacier mass loss in the early twenty-first century. *Nature*, 592(7856), 726-731.
- Immerzeel, W. W., Wanders, N., Lutz, A. F., Shea, J. M., & Bierkens, M. F. P. (2015). Reconciling high-altitude precipitation in the upper Indus basin with glacier mass balances and runoff. *Hydrology and Earth System Sciences*, 19(11), 4673-4687.
- Koutroulis, A. G., Papadimitriou, L. V., Grillakis, M. G., Tsanis, I. K., Warren, R., & Betts, R. A. (2019). Global water availability under high-end climate change: A vulnerability based assessment. *Global and Planetary Change*, 175, 52-63.
- Liaqat, M. U., Grossi, G., & Ranzi, R. (2022). Characterization of interannual and seasonal variability of hydro-climatic trends in the Upper Indus Basin. *Theoretical and Applied Climatology*, 147(3), 1163-1184.

- Lutz AF, Immerzeel WW, Shrestha AB, Bierkens MFP. Consistent increase in High Asia's runoff due to increasing glacier melt and precipitation. *Nature Climate Change* 4, 587-592 (2014).
- Lutz, A. F., Immerzeel, W. W., Kraaijenbrink, P. D., Shrestha, A. B., & Bierkens, M. F. (2016). Climate change impacts on the upper Indus hydrology: sources, shifts and extremes. *PloS one*, 11(11), e0165630.
- Marzeion, B., Hock, R., Anderson, B., Bliss, A., Champollion, N., Fujita, K., ... & Zekollari, H. (2020). Partitioning the uncertainty of ensemble projections of global glacier mass change. *Earth's Future*, 8(7), e2019EF001470.
- Millan, R., Mouginot, J., Rabatel, A., & Morlighem, M. (2022). Ice velocity and thickness of the world's glaciers. *Nature Geoscience*, 15(2), 124-129.
- Sakai, A., & Fujita, K. (2017). Contrasting glacier responses to recent climate change in high-mountain Asia. *Scientific reports*, 7(1), 1-8.
- Schewe, J., Heinke, J., Gerten, D., Haddeland, I., Arnell, N. W., Clark, D. B., ... & Kabat, P. (2014). Multimodel assessment of water scarcity under climate change. *Proceedings of the National Academy of Sciences*, 111(9), 3245-3250.
- Shafeeque, M., Luo, Y., Wang, X., & Sun, L. (2019). Revealing vertical distribution of precipitation in the glacierized upper Indus basin based on multiple datasets. *Journal of Hydrometeorology*, 20(12), 2291-2314.
- Shi, P., Duan, K., Nicholson, K. N., Han, B., Klaus, N., & Yang, J. (2020). Modeling past and future variation of glaciers in the Dongkemadi Ice Field on central Tibetan Plateau from 1989 to 2050. *Arctic, Antarctic, and Alpine Research*, 52(1), 191-209.
- Thompson, J. R., Gosling, S. N., Zaherpour, J., & Laizé, C. L. R. (2021). Increasing risk of ecological change to major rivers of the world with global warming. *Earth's Future*, 9(11), e2021EF002048.
- WGMS: Fluctuations of Glaciers Database. World Glacier Monitoring Service, Zurich, Switzerland. DOI:10.5904/wgms-fog-2021-05. Online access: <http://dx.doi.org/10.5904/wgms-fog-2021-05> (2021).
- Zhang, Q., Chen, Y., Li, Z., Xiang, Y., Li, Y., & Sun, C. (2022). Recent Changes in Glaciers in the Northern Tien Shan, Central Asia. *Remote Sensing*, 14(12), 2878.